# Sepsis impedes EAE disease development and diminishes autoantigen-specific naive CD4 T cells

Isaac J Jensen[1†], Samantha N Jensen[1†], Frances V Sjaastad[2],
Katherine N Gibson-Corley[3], Thamothrampillai Dileepan[4], Thomas S Griffith[5],
Ashutosh K Mangalam[6*], Vladimir P Badovinac[7*]

[1]Interdisciplinary Graduate Program in Immunology, University of Iowa, Iowa City, United States; [2]Microbiology, Immunology, and Cancer Biology PhD Program, University of Minnesota, Minneapolis, United States; [3]Department of Pathology, University of Iowa, Holden Comprehensive Cancer Center, University of Iowa Hospitals and Clinics, Iowa City, United States; [4]Department of Microbiology and Immunology, University of Minnesota, Center for Immunology, Minneapolis, United States; [5]Microbiology, Immunology, and Cancer Biology PhD Program, Department of Urology, Center for Immunology, Minneapolis VA Health Care System, University of Minnesota, Minneapolis, United States; [6]Interdisciplinary Graduate Program in Immunology, Department of Pathology, University of Iowa, Iowa City, United States; [7]Interdisciplinary Graduate Program in Immunology, Department of Pathology, Department of Microbiology and Immunology, University of Iowa, Iowa City, United States

*For correspondence:
ashutosh-mangalam@uiowa.edu
(AKM);
vladimir-badovinac@uiowa.edu
(VPB)

†These authors contributed equally to this work

Competing interests: The authors declare that no competing interests exist.

**Abstract** Evaluation of sepsis-induced immunoparalysis has highlighted how decreased lymphocyte number/function contribute to worsened infection/cancer. Yet, an interesting contrast exists with autoimmune disease development, wherein diminishing pathogenic effectors may benefit the post-septic host. Within this framework, the impact of cecal ligation and puncture (CLP)-induced sepsis on the development of experimental autoimmune encephalomyelitis (EAE) was explored. Notably, CLP mice have delayed onset and reduced disease severity, relative to sham mice. Reduction in disease severity was associated with reduced number, but not function, of autoantigen (MOG)-specific pathogenic CD4 T cells in the CNS during disease and draining lymph node during priming. Numerical deficits of CD4 T cell effectors are associated with the loss of MOG-specific naive precursors. Critically, transfer of MOG-TCR transgenic (2D2) CD4 T cells after, but not before, CLP led to EAE disease equivalent to sham mice. Thus, broad impairment of antigenic responses, including autoantigens, is a hallmark of sepsis-induced immunoparalysis.

## Introduction

Sepsis poses both a significant health concern, affecting 1.7 million and killing 270,000 Americans yearly, and economic burden (> $20 billion annually) (*CDC, 2020*). Sepsis is characterized by cytokine storm, a maladaptive response to a mismanaged infection, that is comprised of both pro- and anti-inflammatory cytokines. Although the acute cytokine storm is a grave situation, patients who survive a septic event are susceptible to further complications with increased susceptibility to unrelated secondary infection, increased viral reactivation, and decreased 5-year survival relative to non-septic patients (*Dombrovskiy et al., 2007*; *Donnelly et al., 2015*; *Gaieski et al., 2013*; *Kutza et al., 1998*; *Walton et al., 2014*). These factors have been attributed to the sepsis-induced lymphopenic

**eLife digest** Sepsis is a life-threatening condition that can happen when the immune system overreacts to an infection and begins to damage tissues and organs in the body. It causes an extreme immune reaction called a cytokine storm, where the body releases uncontrolled levels of cytokines, proteins that are involved in coordinating the body's response to infections. This in turn activates more immune cells, resulting in hyperinflammation.

People who survive sepsis may have long-lasing impairments in their immune system that may leave them more vulnerable to infections or cancer. But scientists do not know exactly what causes these lasting immune problems or how to treat them.

The fact that people are susceptible to cancer and infection after sepsis may offer a clue. It may suggest that the immune system is not able to attack bacteria or cancer cells. One way to explore this clue would be to test the effects of sepsis on autoimmune diseases, which cause the immune system to attack the body's own cells. For example, in the autoimmune disease multiple sclerosis, the immune system attacks and destroys cells in the nervous system. If autoimmune disease is reduced after sepsis, it would suggest the cell-destroying abilities of the immune system are lessened.

Using this approach, Jensen, Jensen et al. show that sepsis reduces the number of certain immune cells, called CD4 T cells, which are are responsible for an autoimmune attack of the central nervous system. In the experiments, mice that survived sepsis were evaluated for their ability to develop a multiple sclerosis-like disease. Mice that survived sepsis developed less severe or no autoimmune disease. After sepsis, these animals also had fewer CD4 T cells. However, when these immune cells were reinstated, the autoimmune disease emerged.

The experiments help explain some of the immune system changes that occur after sepsis. Jensen, Jensen et al. suggest that rather than being completely detrimental, these changes may help to block harmful autoimmune responses. The experiments may also hint at new ways to combat autoimmune diseases by trying to replicate some of the immune-suppressing effects of sepsis. Studying the effect of sepsis on other autoimmune diseases in mice might provide more clues.

state and functional deficits of the surviving cells (*Hotchkiss et al., 2006*; *Hotchkiss et al., 2005*; *Hotchkiss et al., 2001*). This state of immunoparalysis is so profound that the majority of sepsis-associated mortality has shifted to be late deaths following secondary infections or other morbid conditions as a result of immunologic impairment (*Delano and Ward, 2016*). Therefore, emphasis has been strongly shifted toward determining the mechanisms by which sepsis impairs the immune response to subsequent infection or cancer (*Chen et al., 2019*; *Condotta et al., 2015*; *Condotta et al., 2013*; *Danahy et al., 2019a*; *Jensen et al., 2018b*). Yet, this focus on the sepsis-induced loss of beneficial host responses has come at the expense of understanding how sepsis may influence other subsequent maladaptive immune responses. As opposed to infection/cancer, wherein effectors promote disease control, autoantigen-specific effectors promote disease and are detrimental to the host.

Multiple sclerosis (MS) is a chronic neuroinflammatory disease of the CNS and is potentially the most common non-traumatic cause of CNS disability in young adults (*Compston and Coles, 2002*). MS poses a major personal and economic burden; it effects mostly women and on average presents at age 30, an age crucial for family planning (*Fox, 2004*). The etiology of MS involves a complex interaction between genetic and environmental components, resulting in a clinical presentation including, but not limited to, sensory, motor, and/or cognitive deficits (*Dendrou et al., 2015*; *Freedman et al., 2018*). MS is thought to result from autoreactive CD4 T cell responses to the myelin antigens in the CNS followed by inflammatory cellular infiltration, demyelination, and neurodegeneration. In experimental autoimmune encephalomyelitis (EAE), a murine model of MS, this role for CD4 T cells is well-established and induced by immunization against myelin antigens (*Stinissen et al., 1997*).

While there is a dearth of knowledge regarding the impact of sepsis on autoimmunity, descriptions of how sepsis influences specific cell subsets can provide insight into how autoimmune disease may be influenced. With regard to the critical role for CD4 T cells in EAE disease, the sepsis-induced

lymphopenic state is known to impact both naive and memory CD4 T cells (*Cabrera-Perez et al.,* *2016*; *Cabrera-Perez et al., 2015*; *Chen et al., 2017*; *Jensen et al., 2018a*; *Sjaastad et al., 2020b*). In particular, sepsis can diminish the number of naive CD4 T cell precursors subsequently limiting the number of cells capable of responding to a given antigen (*Cabrera-Perez et al., 2015*; *Martin et al., 2020*). Additionally, sepsis can impair the effector function of surviving T cells further limiting the capacity of cells to mount an antigen-specific response (*Mohr et al., 2012*; *Pötschke et al., 2013*; *Sjaastad et al., 2018*). Sepsis can also impair antigen-specific T cell responses through various cell extrinsic factors, including diminished number/function of dendritic cells (DCs) and diminished capacity of endothelial cells to promote chemotaxis (*Danahy et al.,* *2017*; *Strother et al., 2016*). Cumulatively, these findings suggest that, while detrimental to control of pathogens/cancer, sepsis may alternately diminish a host's capacity to develop autoimmunity.

Herein, we demonstrate that following cecal ligation and puncture (CLP), mice have diminished EAE disease severity. Autoantigen-specific effector CD4 T cells were reduced in CLP hosts with EAE due to a loss of their naive precursors, which was the determining factor in the impediment of EAE disease. These data further define how the sepsis-induced immunoparalysis reduces host capacity to generate antigen-specific responses, regardless of whether it is a foreign antigen or an endogenous autoantigen.

## Results

### Sepsis reduces EAE disease severity and the number of CNS infiltrating pathogenic effector CD4 T cells

The influence of the immunoparalysis phase of sepsis, which follows the resolution of the cytokine storm, on the ability of the host to develop autoimmunity is undefined. To address this relationship, we employed well-established models of polymicrobial sepsis and inducible autoimmunity, CLP and EAE, respectively. CLP effectively mimics the pathophysiology of acute peritonitis and corresponding septic inflammation, with the capacity to modulate disease severity through adjustment of surgical parameters (*Dejager et al., 2011*; *Sjaastad et al., 2020a*). Likewise, EAE presents an ideal model system to study the impact of sepsis on the development of autoimmune disease since it is a robust, inducible system allowing for manipulations prior to disease initiation (*Stinissen et al.,* *1997*).

CLP, sham, and non-surgery mice were immunized with MOG$_{35-55}$ peptide 5 days post-surgery, a time when the cytokine storm has resolved (*Danahy et al., 2019b*), to induce EAE and disease progression was monitored (*Figure 1a*). Of note: unless otherwise stated, experimental timepoints are defined with respect to EAE immunization. Importantly, while sham and no surgery control groups of mice developed robust EAE clinical disease, CLP survivors (day 5 CLP group) had substantially diminished disease severity (*Figure 1b,c*) and delayed disease onset (*Figure 1d*). Further, CLP hosts do not achieve similar disease severity even when disease course is prolonged (40 days) (*Figure 1—figure supplement 1*) indicating that diminished disease is not due to delayed disease onset.

Sepsis can lead to long-term functional deficits by immune cells (*Jensen et al., 2018a*). Therefore, to address whether the CLP-induced reduction in EAE disease was a durable phenomenon, we included a group of mice that had undergone CLP surgery 25 days prior to EAE immunization and evaluated the development of EAE disease (*Figure 1a*). Interestingly, even at this later time point CLP survivors continued to exhibit reduced disease severity (*Figure 1b,c*) and delayed disease onset (*Figure 1d*) relative to sham or no surgery controls, though disease severity was increased relative to day 5 CLP mice. The more severe disease relative to day 5 mice was due to a noted variability among the day 25 CLP mice, wherein some individuals developed severe disease and others had minimal disease. This difference in EAE severity suggests that the influence of the immunoparalysis state may wain with time, at least in some of the mice. Therefore, to capture the maximal state of suppression, we focused on comparisons of day five sham and CLP hosts.

It is important to stress that sham surgery is the preferred control for assessing how the septic event influences the development of subsequent T-cell-mediated immune responses, since those mice experience the same surgical procedures as CLP mice (excluding those that initiate the septic event) (*Cuenca et al., 2010*; *Sjaastad et al., 2020a*). In particular, anesthesia, such as the ketamine used for the surgical procedures, can influence CD4 T cell function and capacity to induce EAE

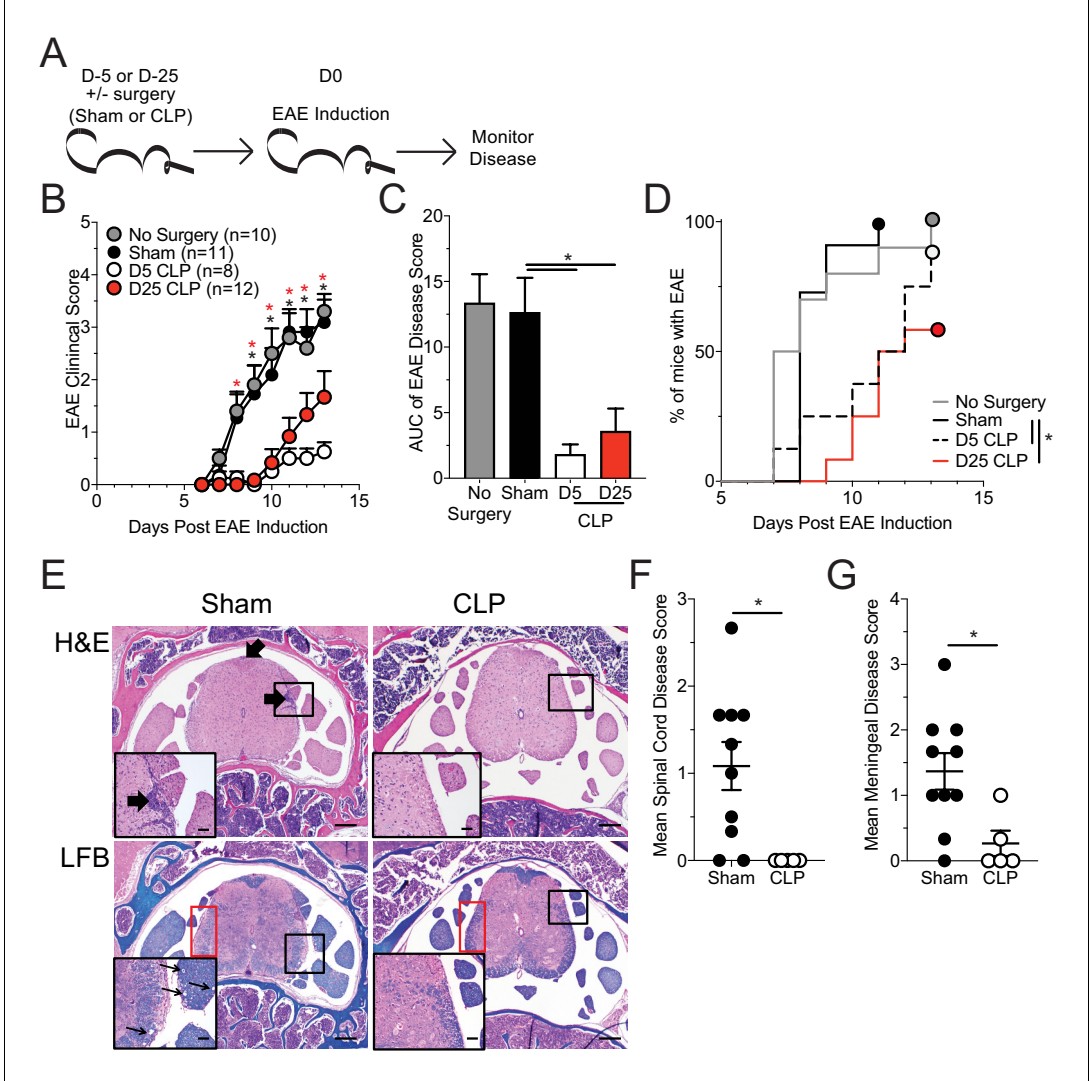

**Figure 1.** Sepsis ablates EAE disease and reduces CNS pathology. (**A**) Experimental design: C57Bl/6 mice were either left naive or underwent sham or CLP surgery. All groups were immunized s.c. with MOG$_{35-55}$ emulsified in complete Freunds adjuvant (CFA) on both flanks 5 days after surgery and given pertussis toxin (PTX) i.p. on the day of immunization as well as 2 days later to induce EAE. An additional group of mice that underwent CLP 25 days prior were also immunized on the same day. Disease onset and severity were monitored over the subsequent 14 days. (**B**) Average EAE disease score following EAE induction for mice that had either no surgery (gray), Sham surgery (black), D5 CLP surgery (white), or D25 CLP surgery (red). (**C**) Area under the curve (AUC) of disease scores in panel B following EAE induction. (**D**) Time to onset of first EAE symptoms following EAE induction in no surgery (gray), Sham surgery (black), D5 CLP surgery (dashed), and D25 CLP (red) mouse groups. (**E**) Representative photomicrographs of the lumbar spine from mice with EAE that underwent either sham or D5 CLP surgery. Thick arrows indicate areas of inflammation that was scored on H and E stained slides. Luxol fast blue (LFB) stains were also performed and showed subtle demyelination and axonal sheath swelling in sham mice (red box). Vacuolation is indicated in the inserts by the thin arrows. Bars = 200 µm (inset bars = 50 µm). Mean H and E inflammation score of the (**F**) spinal cord and (**G**) meninges of mice 15 days-post EAE induction. Data are representative from three independent experiments with 8–12 mice per group. D25 CLP data are from a single experiment. *p<0.05. Error bars represent the standard error of the mean.

The online version of this article includes the following source data and figure supplement(s) for figure 1:

**Source data 1.** Source data for *Figure 1B–D*.

**Source data 2.** Source data for *Figure 1E*.

**Source data 3.** Source data for *Figure 1F,G*.

**Figure supplement 1.** Sepsis ablates EAE disease and reduces CNS pathology.

**Figure supplement 1—source data 1.** Source data for *Figure 1—figure supplement 1*.

**Figure supplement 2.** Altered systemic cytokine response to EAE immunization in septic mice.

**Figure supplement 2—source data 1.** Source data for *Figure 1—figure supplement 2*.

*Figure 1 continued on next page*

*Figure 1 continued*

**Figure supplement 3.** Reduced representation and number of activated microglia, monocytes, and macrophages in the CNS of CLP mice following EAE immunization.

**Figure supplement 3—source data 1.** Source data for *Figure 1—figure supplement 2*.

(*Hou et al., 2016*; *Lee et al., 2017*; *Ohta et al., 2009*). Thus, to ensure that differences observed were due to the septic event, rather than being imposed by surgery, we utilized sham surgery as a relevant control to CLP procedure for the remainder of our experiments.

To further address how sepsis altered the systemic parameters of disease, we assessed serum cytokine/chemokine concentrations in sham and CLP hosts both prior (day 0) to and during EAE disease (day 15 post-EAE induction). Notably, the differences in clinical scores between sham and CLP hosts were associated with an altered pattern of expression by various systemic cytokines following EAE immunization (*Figure 1—figure supplement 2a*); this included numerous effector cytokines associated with EAE disease development (*Figure 1—figure supplement 2b–i*). Given the difference observed in disease between sham and CLP mice, in conjunction with an altered cytokine response, histopathologic scoring of the parenchymal and meningeal regions of the spinal cord during active disease was performed. Corresponding with their milder disease, CLP mice with EAE had less CNS pathology as highlighted by reduced inflammatory cell infiltrate (*Figure 1e,H and E*) in both the parenchyma (*Figure 1f*) and meninges (*Figure 1g*) of the spinal cord, the primary target of EAE inflammation. Further, CLP mice had milder axonal demyelination (*Figure 1e*, LFB) and vacuole formation, factors associated with EAE-induced paralysis.

The difference in infiltrate observed extended to a reduced frequency and number of both microglia and infiltrating monocytes and macrophages present within the CNS in CLP hosts, relative to their sham counterparts (*Figure 1—figure supplement 3a–e*). A significant reduction in MHC II expression was also observed on both microglia and infiltrating monocytes/macrophages in CLP hosts compared to sham-treated mice (*Figure 1—figure supplement 3f,h*). This reduction compounded with the already reduced number of myeloid cells leading to ~100 fold reduction in MHC II-expressing cells in CLP hosts (*Figure 1—figure supplement 3g,i*). Prior studies have demonstrated alterations in MHC expression by myeloid cells which are dysregulated following sepsis (*Jensen et al., 2020*; *Monneret et al., 2006*; *Siegler et al., 2018*). MHC II expression is not only a marker of activation of these cell populations but also serves to present antigen to CD4 T cells, including autoreactive T cells that are critical mediators of disease in EAE. Therefore, this global infiltrate reduction in CLP hosts may reflect the lack of an autoimmune response in the CNS for which CD4 T cells are critical mediators.

To address whether the reduction in histological inflammatory cell infiltrate reflected a difference in the accumulation of pathogenic antigen-specific CD4 T cells, total CNS (brain and spinal cord) was harvested from sham and CLP mice, following perfusion, at day 15 post-EAE induction (*Figure 2a*). Flow cytometric analysis of the CNS revealed a decreased frequency (*Figure 2b,c*) and number (*Figure 2d*) of antigen-experienced CD4$^+$ CD11a$^+$ cells in the CNS (*Christiaansen et al., 2017*). Further, tetramer staining of CD4 T cells reactive to the MOG antigen revealed a further reduction in the frequency (*Figure 2e,f*) and number (*Figure 2g*) of this pathogenic effector population.

These data suggest that the reduction in disease severity was potentially due to lack of autoantigen-specific CD4 T cells infiltrating the CNS; however, sepsis is also known to influence T cell function. Thus, direct ex vivo analysis of the MOG-specific CD4 T cells in the CNS was evaluated by intracellular cytokine staining at the same timepoint. No differences were observed in the frequencies of MOG-specific CD4 T cells producing pathogenic effector cytokines IFNγ, IL-17A, and TNFα without further stimulation directly ex vivo (*Figure 3a,b*; *Figure 3—figure supplement 1*; *Lee et al., 2019*). However, the numerical deficit in MOG-specific CD4 T cells led to a significant reduction in the number of IFNγ- and TNFα-producing cells with a trending decrease in IL-17A-producing cells (*Figure 3c*). This finding indicates the autoantigen-specific CD4 T cells present in the CNS after CLP surgery are functionally competent and potentially were the least impacted by the septic event. In summary, the data presented so far suggest that the numerical deficit in CNS-infiltrating pathogenic effectors might stem from insufficient priming and/or expansion of encephalitogenic CD4 T cells in the draining lymph node in the periphery.

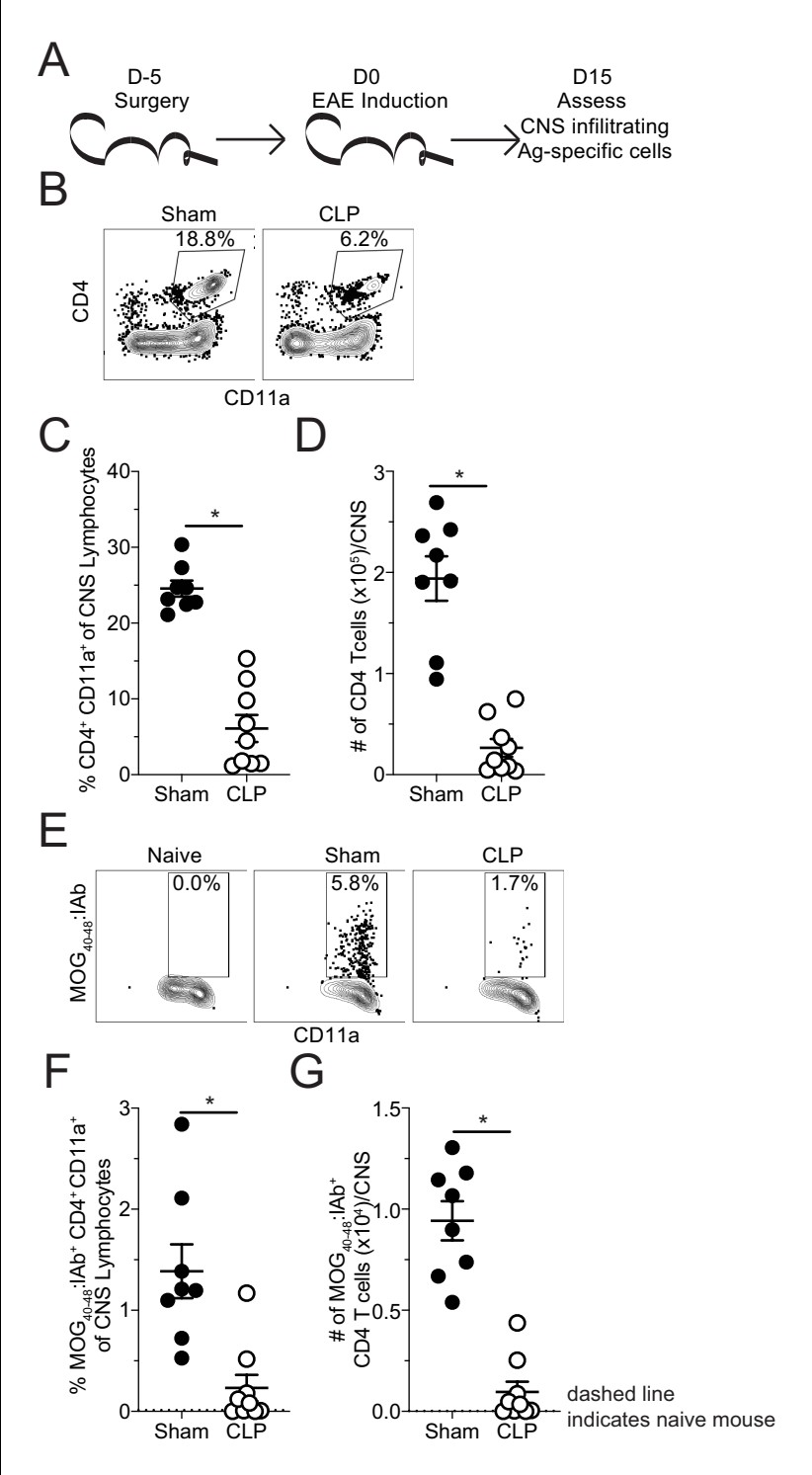

**Figure 2.** Fewer MOG-specific CD4 T cells are present in the CNS of CLP mice. (**A**) Experimental design: C57Bl/6 mice underwent either sham or CLP surgery. EAE induction occurred 5 days after surgery. Mice were perfused 15 days post-EAE induction and CNS was harvested. (**B**) Representative flow plots for CNS CD4 T cells, gated on lymphocytes, from sham and CLP mice. (**C**) Frequency and (**D**) number of CNS CD4 T cells in sham and CLP mice. (**E**) Representative flow plots for CNS MOG-specific CD4 T cells, gated on total CD4 T cells, from Naive (staining control), sham, and CLP mice. (**F**) Frequency and (**G**) number of CNS MOG-specific CD4 T cells in sham and CLP mice. Dashed line indicates staining and number from naive control mouse. Data are representative from two independent experiments with 8–10 mice per group. *p<0.05. Error bars represent the standard error of the mean.

*Figure 2 continued on next page*

*Figure 2 continued*

The online version of this article includes the following source data for figure 2:

**Source data 1.** Source data for *Figure 2*.

## Sepsis-induced loss of naive CD4 T cells is associated with a reduced number of autoantigen-specific CD4 T cell precursors

To address whether the CLP hosts had a priming deficit for autoantigen-specific CD4 T cells, MOG-specific CD4 T cells were evaluated in the draining inguinal lymph node (iLN), 7 days after EAE induction (*Figure 4a*). This is a time during which CD4 T cells are being primed and expanding that precedes the development of clinical disease (*Bischof et al., 2004*). Similar to the CNS during peak disease, both the frequency (*Figure 4b*) and number (*Figure 4c*) of MOG-specific CD4 T cells were decreased in the iLN of CLP hosts.

Since lymphopenia is a hallmark of sepsis and differentially affects CD4 T cell precursor populations (*Cabrera-Perez et al., 2015*), a reduction in precursor numbers is one possibility for the

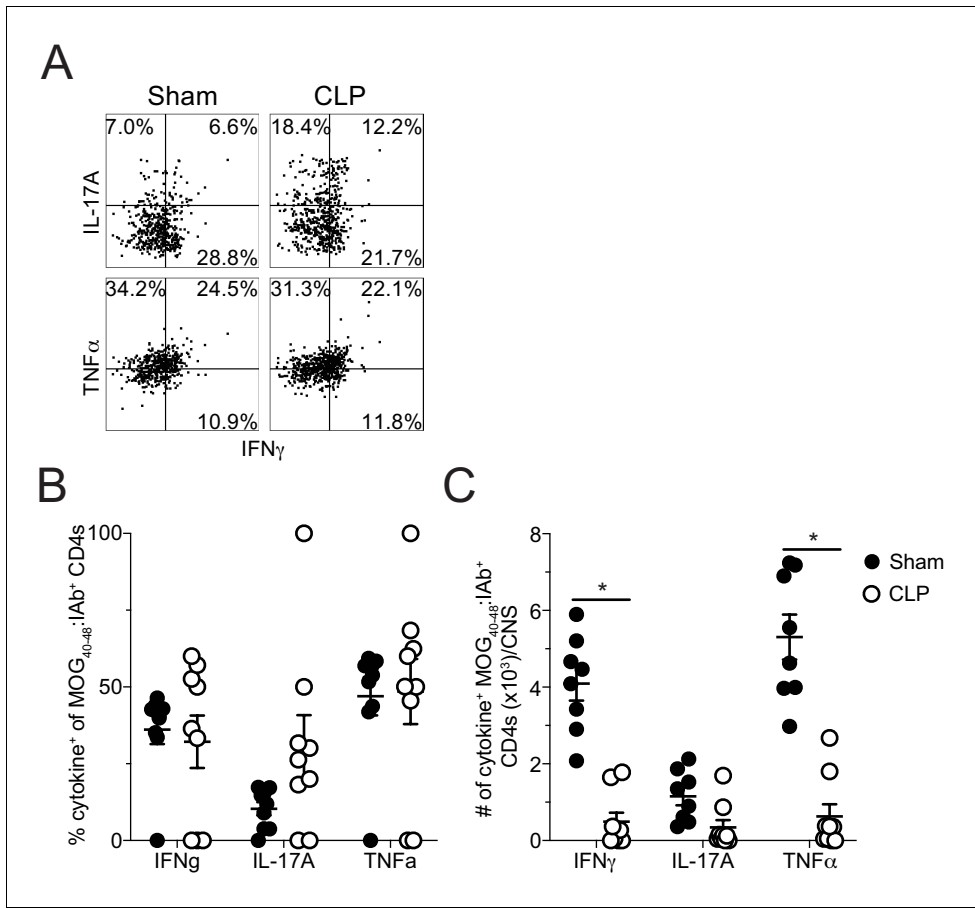

**Figure 3.** Sepsis reduces the number of cytokine-producing MOG-specific CD4 T cells in the CNS. (**A**) Representative flow plots of direct ex vivo IFNγ, IL-17A, and TNFα producing cells, gated on MOG-specific CD4 T cells, from sham (left) and CLP (right) mice. Numbers indicate frequency of cytokine-positive cells per quadrant. (**B**) Frequency and (**C**) number of IFNγ-, IL-17A-, and TNFα-producing MOG-specific CD4 T cells in the CNS of sham and CLP mice. Data are representative from two independent experiments with 8–10 mice per group. *p<0.05. Error bars represent the standard error of the mean.

The online version of this article includes the following source data and figure supplement(s) for figure 3:

**Source data 1.** Source data for *Figure 3*.

**Figure supplement 1.** FMO controls for direct ex vivo cytokine staining.

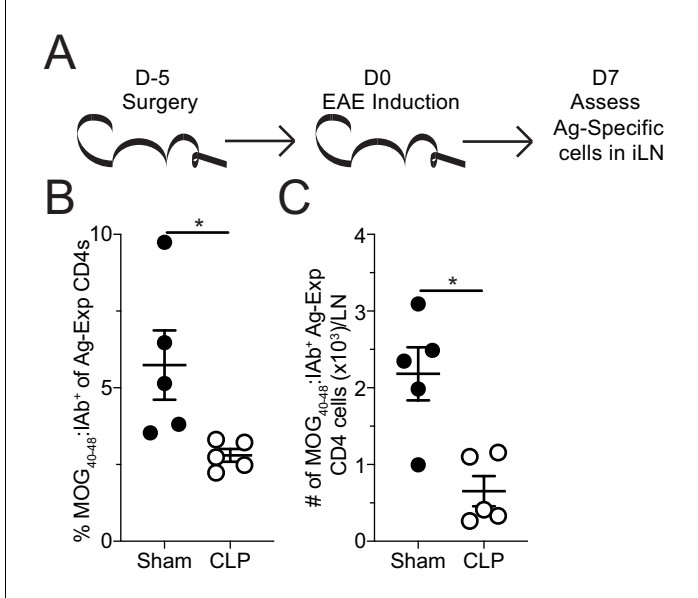

**Figure 4.** Sepsis reduces the number MOG-specific CD4 T cells present in sites of T cell priming following EAE induction. (**A**) Experimental design: C57Bl/6 mice underwent sham or CLP surgery. EAE induction occurred 5 days after surgery. Draining inguinal lymph nodes (iLN) were harvested 7 days after EAE disease induction. (**B**) Frequency and (**C**) number of MOG-specific CD4 T cells in the iLN of sham and CLP mice. Data are representative from three independent experiments with five mice per group. *p<0.05. Error bars represent the standard error of the mean.

The online version of this article includes the following source data for figure 4:

**Source data 1.** Source data for *Figure 4*.

decreased number of autoantigen-specific CD4 T cells in the iLN at the day 7 priming timepoint. Thus, the precursor frequency of MOG-specific naive CD4 T cells was interrogated through tetramer enrichment of splenic CD4 T cells 5 days after surgery (i.e. the day of EAE induction, *Figure 5a*). This analysis revealed a significant numerical loss of MOG-specific CD4 T cell precursors in CLP hosts. These data suggest CLP-induced lymphopenia may protect against the development of EAE by reducing the number of naive autoantigen-specific CD4 T cells. The subsequent loss of these naive cells would explain the corresponding reduction in the number of effector cells present in the both the lymph node and CNS following immunization.

To address whether the prior observation, that CLP hosts have a lasting decrease in the severity of EAE disease (*Figure 1b–d*), is also associated with a lasting numerical deficit, the number of MOG-specific CD4 T cells was enumerated in sham and CLP hosts 25 days post-surgery. Consistent with our observation of disparate disease development among CLP hosts, we observed a corresponding bifurcation in the number of MOG-specific CD4 T cells in individual mice (*Figure 5d*) further suggesting that the ability to initiate EAE disease might correspond to the number of MOG-specific CD4 T cells available. Additionally, similar to prior reports (*Cabrera-Perez et al., 2016*; *Jensen et al., 2018a*; *Skirecki et al., 2020*), an increased frequency of activated (CD44hi) CD4 T cells was observed in CLP hosts (*Figure 5e*). Importantly, this extended to an increase in the frequency of memory-like CD44hi cells among MOG-specific CD4 T cells in CLP hosts (*Figure 5f*). This increase in CD44 expression could occur either be through homeostatic proliferation following the severe lymphodepletion that occurs following a septic event or potentially through antigen release due to sepsis-induced tissue damage (*Cabrera-Perez et al., 2016*; *Cabrera-Perez et al., 2015*). Thus, even for those cells that do recover in numbers they are phenotypically and functionally distinct, which may further contribute to the differences in EAE disease development and progression following immunization.

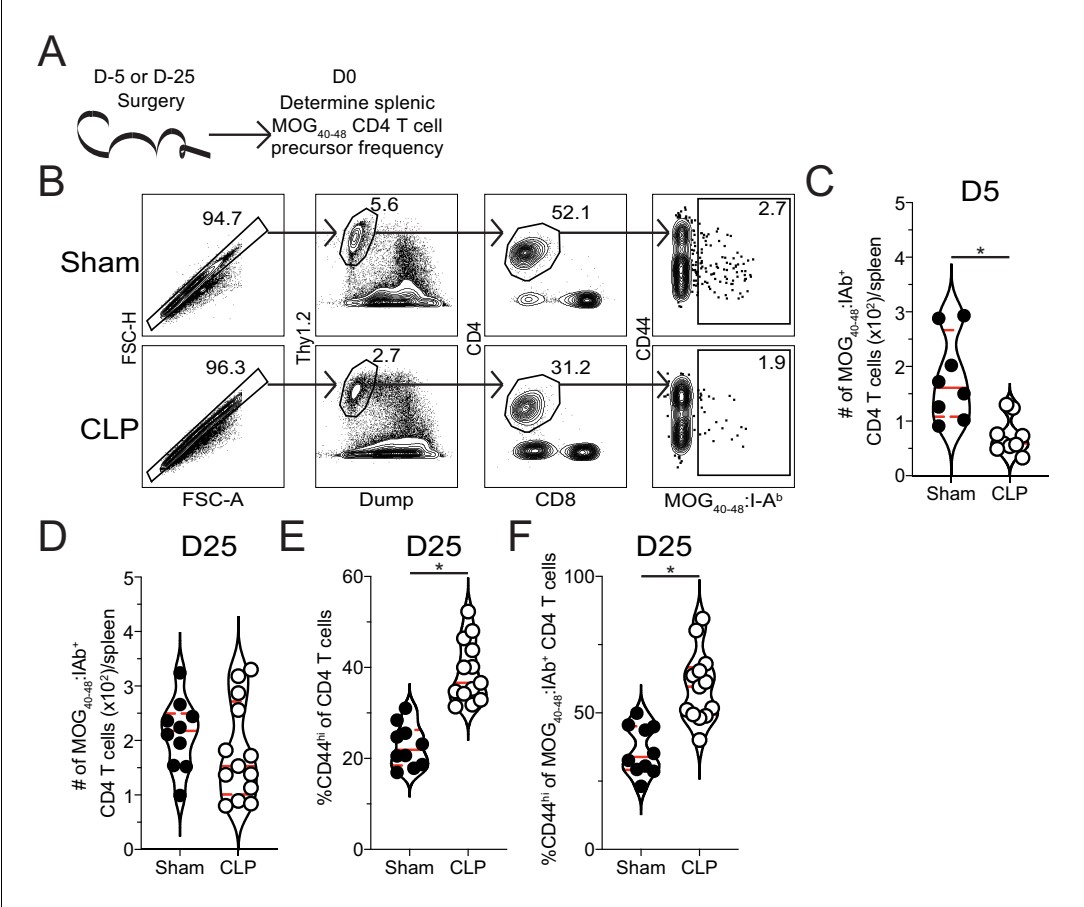

**Figure 5.** Sepsis reduces the number of naive MOG-specific CD4 T cell precursors. (**A**) Experimental design: C57Bl/6 mice underwent sham or CLP surgery. MOG-specific CD4 T cell precursors were enriched from the spleen 5 days after surgery via tetramer pulldown and enumerated. (**B**) Representative gating strategy for identifying MOG-specific CD4 T cell precursors from sham and CLP mice. Number of splenic MOG-specific CD4 T cell precursors in sham and CLP mice (**B**) 5 and (**C**) 25 days post-surgery. Frequency of antigen-experienced (CD44hi) (**E**) total and (**F**) MOG-specific CD4 T cells. Data are combined from two independent experiments with 8–13 mice per group. *p<0.05. Error bars represent the standard error of the mean. The online version of this article includes the following source data for figure 5:

**Source data 1.** Source data for *Figure 5*.

## Loss of autoantigen-specific naive CD4 T cells is the determining factor in sepsis-induced impediment of EAE

To further delineate the contributions of the sepsis influence directly on CD4 T cells or their environment, we utilized an adoptive transfer system wherein naive MOG-specific, TCR-transgenic 2D2 CD4 T cells were transferred into congenically distinct recipient mice. By having the fixed TCR of the 2D2 cells, the function of the 2D2 cells could be equally assessed between sham and CLP hosts. Mice receiving 2D2 cells 1 day prior to surgery (6 days prior to EAE induction) were part of the 'pre-transfer' group. Alternatively, a second cohort of mice received 2D2 cells 4 days post-surgery (1 day prior to EAE induction), a time at which sepsis-associated inflammation has resolved and the immunoparalysis state is established (*Danahy et al., 2019b*), were in the 'post-transfer' group. 2D2 cells in the pre-transfer group are exposed to both the intrinsic and extrinsic changes that sepsis induces. In contrast, the 2D2 cells in the post-transfer experiments were only influenced by the sepsis-induced changes in the environment, such as reduced DC function (*Strother et al., 2016*), reduced capacity to traffic across endothelium (*Danahy et al., 2017*), and alterations in the proportion of regulatory CD4 and CD8 T cells (*Sharma et al., 2015*; *Sinha et al., 2015*). These impairments are all likely to be present in the context of post-septic environment and may influence the development of EAE.

Thus, the post-septic environmental factors which influence CD4 T cells can be assessed by transferring cells into the host after the immunoparalysis state has been established. Further, by comparing the pre- and post-transfer cells from sham and CLP mice the intrinsic and extrinsic influences of sepsis can be parsed.

Thus, to address the intrinsic and extrinsic impact of sepsis on priming of CD4 T cells, 2D2 cell number, proliferation, as assessed by recent proliferation marker Ki67 (*Miller et al., 2018*), and apoptosis, as assessed by presence of active caspase 3/7 (FLICA[+]) and membrane depolarization (PI[+]), were evaluated in the iLN 7 days post-EAE induction (*Figure 6a,b*). Notably, cells that were transferred prior to surgery were numerically diminished in CLP hosts relative to sham counterparts (*Figure 6c*) recapitulating the observations with endogenous MOG-specific CD4 T cells (see *Figure 5*). Strikingly, 2D2 cells that were transferred after surgery were numerically equivalent between sham and CLP hosts, suggesting the septic environment (e.g. diminished DC number/function, increased proportion of regulatory CD8 and/or CD4 T cells) did not limit their capacity to expand

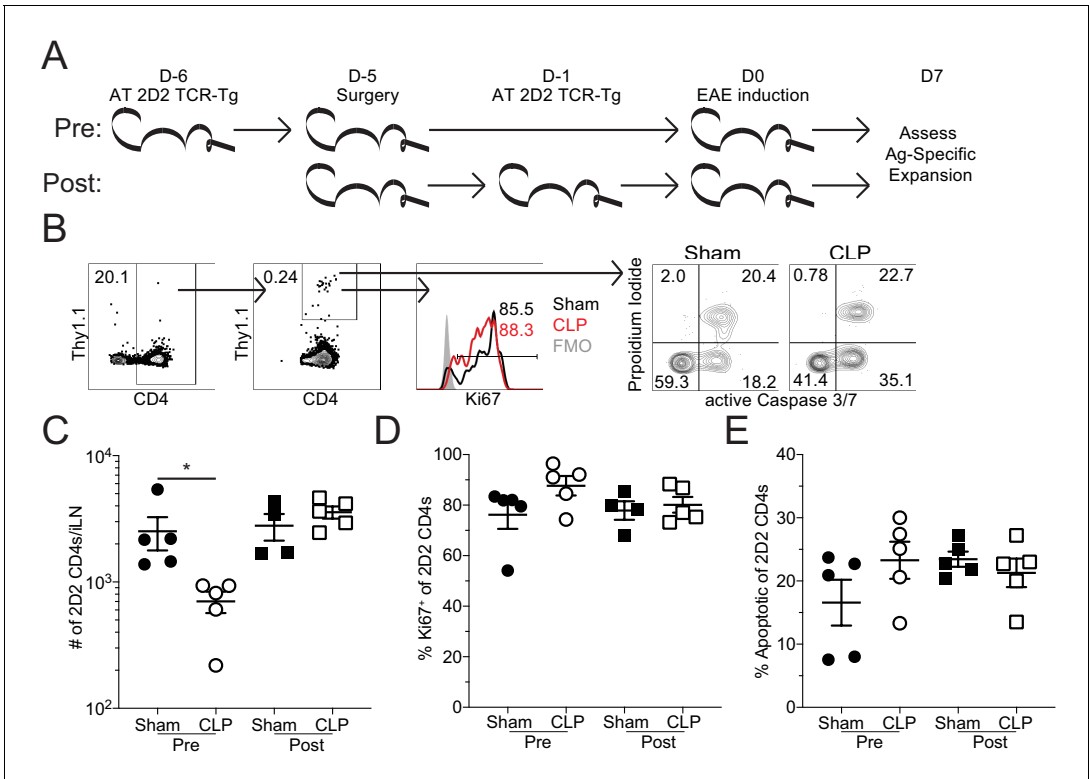

**Figure 6.** Sepsis reduces the number of MOG-specific CD4 T cells but not their capacity to proliferate. (**A**) Experimental design: Thy1.2 C57Bl/6 mice were separated into Pre- and Post-transfer groups. The Pre-transfer group received $5 \times 10^3$ naive Thy1.1 2D2 TCR-Tg CD4 T cells 1 day before sham or CLP surgery. The Post-transfer group underwent sham or CLP surgery then received $5 \times 10^3$ naive Thy1.1 2D2 TCR-Tg CD4 T cells 4 days later. EAE was induced in both the Pre- and Post-transfer groups 5 days after surgery (Pre: 6 days post 2D2 T cell transfer; Post: 1 day post 2D2 T cell transfer). iLN were harvested 7 days after the transfer. (**B**) Representative gating strategy for identifying transferred 2D2 TCR-Tg CD4 T cells, their expression of the proliferation marker Ki67, and markers of apoptosis (activated caspase3/7 with propidium iodide) from sham and CLP mice. (**C**) Number of transferred 2D2 TCR-Tg CD4 T cells in the iLN of sham and CLP mice. (**D**) Frequency of 2D2 TCR-Tg CD4 T cells expressing Ki67. (**D**) Frequency of apoptotic (FLICA[+] PI[+]) 2D2 TCR-Tg CD4 T cells. Data are representative from two independent experiments with four to five mice per group. *p<0.05. Error bars represent the standard error of the mean.

The online version of this article includes the following source data and figure supplement(s) for figure 6:

**Source data 1.** Source data for *Figure 6B–D*.
**Source data 2.** Source data for *Figure 6B,E*.
**Figure supplement 1.** Sepsis does not influence the expression of Fas, FasL, and TRAIL by autoantigen-specific T cells.
**Figure supplement 1—source data 1.** Source data for *Figure 6—figure supplement 1*.
**Figure supplement 2.** Sepsis does not influence the expression of RORγT, Tbet, and FoxP3 by autoantigen-specific T cells.
**Figure supplement 2—source data 1.** Source data for *Figure 6—figure supplement 2*.

(*Figure 6c*). Further, the transferred 2D2 cells from all groups had a similar proportion of recently proliferated as well as apoptotic (FLICA$^+$ PI$^+$) cells (*Figure 6d,e*). Given that the post-transfer groups had equivalent cell numbers, proliferation, and apoptosis; these data demonstrate that autoantigen CD4 T cells are not limited by the post-septic environment in their capacity to be primed and expand. Thus, while sepsis notably induces a variety of T-cell-extrinsic immunologic impairments (*Danahy et al., 2017*; *Sharma et al., 2015*; *Sinha et al., 2015*; *Strother et al., 2016*), these impairments can be overcome in the context of EAE immunization. In addition, there was no observed difference in the expression of several canonical extrinsic death-inducing proteins known to limit the expansion of CD4 T cells (i.e. Fas, FasL, TRAIL) between sham and CLP hosts for either the pre- or post-transfer groups (*Figure 6—figure supplement 1*) further supporting the notion that survival differences do not account for the numerical deficit in CD4 T cells exposed to sepsis. Therefore, pre-transfer 2D2 cells proliferate and die equivalently in both sham and CLP hosts, indicating that sepsis does not intrinsically impair the capacity of these cells to proliferate in response to cognate antigen recognition.

To directly compare the survival and expansion potential of autoantigen-specific CD4 T cells, congenically distinct Thy1.1 2D2 mice underwent either sham or CLP surgery. The same number of 2D2 cells from sham and CLP mice (1:1 mix) were then adoptively transferred into naive Thy1.2 C57Bl/6-recipient mice. A day later, recipients were either left unimmunized or immunized to induce EAE to address the survival and expansion potential of both subsets of 2D2 CD4 T cells, respectively. Survival potential was assessed in the lymph node 5 days post-transfer, while expansion was assessed at

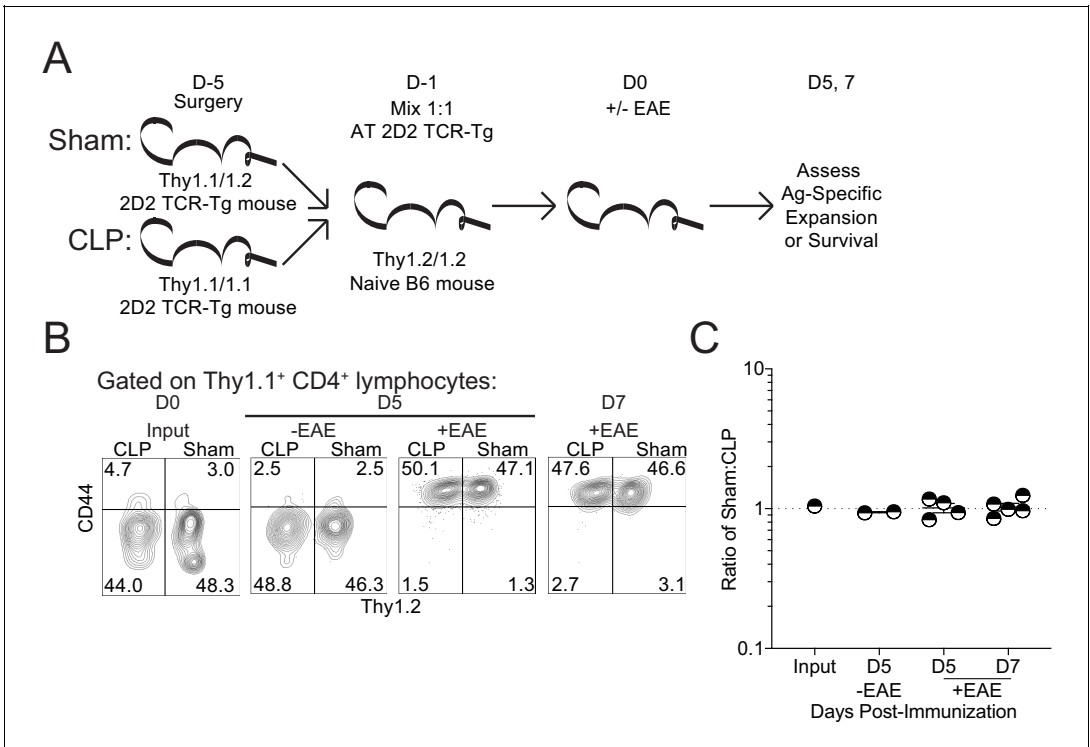

**Figure 7.** Sepsis does not lead to a cell intrinsic deficit in CD4 T cell expansion. (**A**) Experimental design: Thy1.1/1.1 2D2 TCR-Tg mice underwent sham surgery while Thy1.1/1.2 2D2 TCR-Tg mice underwent CLP surgery. 2D2 mice were euthanized 4 days post-surgery and splenic 2D2 cells were mixed at a 1:1 ratio prior to transfer into naive Thy1.2/1.2 B6 mice. Non-EAE mice received $5 \times 10^6$ of each 2D2 population, whereas EAE mice received $5 \times 10^3$ of each 2D2 population. A day after transfer, EAE was either induced or not in the respective recipient group. Survival of the transferred 2D2 cells (assessed in non-EAE hosts) was assessed in the iLN 5 days after transfer. Expansion of the transferred 2D2 cells (assessed in EAE hosts) was assessed in the iLN 5 and 7 days after transfer. (**B**) Representative profiles of the 2D2 input and output on indicated days for both EAE and non-EAE hosts. (**C**) The ratio of Sham to CLP 2D2 cells in the input and output at indicated days for both EAE and non-EAE hosts. Data are from 1 experiment with two to five mice per group. *p<0.05. Error bars represent the standard error of the mean.

The online version of this article includes the following source data for figure 7:

**Source data 1.** Source data for *Figure 7*.

both 5- and 7 days post-transfer (*Figure 7a*). Importantly, no difference was observed in either the survival or expansion potential of 2D2 cells acquired from CLP mice relative to those acquired from sham controls (*Figure 7b,c*). Thus, these data demonstrate that CLP does not change the capacity of autoantigen-specific CD4 T cells to exert their effector function(s) upon transfer into a non-septic environment.

Given the aforementioned differences in the systemic cytokine milieu between sham and CLP hosts following immunization (*Figure 1—figure supplement 1*), it is possible that autoantigen-specific CD4 T cells may be unable to form the relevant effector cell populations of Th1, Th17, and Treg cells known to influence EAE disease development (*Dendrou et al., 2015*). To address this possibility, pre- and post-transfer groups were generated as in *Figure 6* and the number of RORγT (Th17), Tbet (Th1), and FoxP3 (Treg) 2D2s were enumerated in the lymph node 7 days post-immunization (*Figure 6—figure supplement 2a*). As shown in *Figure 6* and *Figure 6—figure supplement 1*, the numerical loss of 2D2 cells during sepsis influenced the number of each of these effector populations within the pre-transfer group but did not impact the number of post-transfer 2D2 cells for each of the effector populations (*Figure 6—figure supplement 2b,c*). Therefore, relative to the pre-transfer group, the post-transfer 2D2s recovered their capacity to develop each of these populations (*Figure 6—figure supplement 2d*). In contrast, endogenous CD4 T cells exposed to septic event remained impaired in their capacity to generate these effector populations (*Figure 6—figure supplement 2e*).

Finally, to determine whether the sepsis-induced loss of MOG-specific CD4 T cell precursors is causal in the reduced disease severity of EAE mice, the same 2D2 transfer and surgery groups were used as before and disease progression was monitored (*Figure 8a*). Coinciding with the cellular expansion seen in the iLN, mice that received 2D2 CD4 T cells prior to surgery developed less severe disease than their sham counterparts (*Figure 8b,c*), similar to results in *Figure 1*. However, CLP mice that received 2D2 CD4 T cells post-surgery developed equivalent disease as sham counterparts (*Figure 8b,c*). This result demonstrates that a determining factor by which sepsis limits EAE stems from a reduction in number of naive MOG-specific CD4 T cell precursors. Importantly, there was a numerical reduction in the CNS infiltration of both transferred 2D2 (*Figure 8d*) and endogenous MOG-specific CD4 T cells (*Figure 8f*) of CLP mice that received transfer prior to surgery demonstrating that both populations of cells were similarly influenced by the septic event. Conversely, there was not a numerical deficit in 2D2 CD4 T cells when transferred into CLP hosts post-surgery (*Figure 8e*), while the number of endogenous MOG-specific CD4 T cells in the CNS in the same hosts were significantly reduced (*Figure 8g*). These data indicate the impact sepsis has on the disease-causing capacity on MOG-specific CD4 T cells, depending on whether the given population was exposed to the septic event. Cumulatively, these data indicate that numerical loss, due to sepsis-induced lymphopenia, of naive autoantigen-specific CD4 T cell precursors is sufficient to explain the protective effect of CLP on EAE disease.

## Discussion

Although the sepsis-induced cytokine storm remains a life-threatening condition, it has also become apparent that the aftermath of the septic event leads to significant changes in the immune systems of individuals who survive. Prior work has established how the sepsis-induced immunoparalysis state negatively impacts a host's subsequent capacity to control infection/cancer. Yet, the consequence of sepsis on other disease states, in particular autoimmunity, remains poorly defined. Herein, we have characterized the septic impact on the development of an autoimmune disease.

Although the impact of sepsis on autoimmunity is understudied, the known influences of sepsis on CD4 T cells and their environment provided a pivotal framework to begin interrogating this relationship (*Cabrera-Perez et al., 2016*; *Cabrera-Perez et al., 2017*; *Cabrera-Perez et al., 2014*; *Cabrera-Perez et al., 2015*; *Chen et al., 2017*; *Hotchkiss et al., 2001*). Our observation of reduced EAE disease severity in CLP mice, associated with the loss of MOG-specific naive CD4 T cell precursors, further extends the characterization of the immunoparalysis state as being non-permissive to the formation of antigen-specific T cell responses. However, the use of cell transfers may have aided in overcoming the T cell extrinsic factors, such as DC number/function and capacity of endothelia to promote cell trafficking. Therefore, these other factors may still be relevant to autoimmune disease development during the sepsis-induced immunoparalysis state.

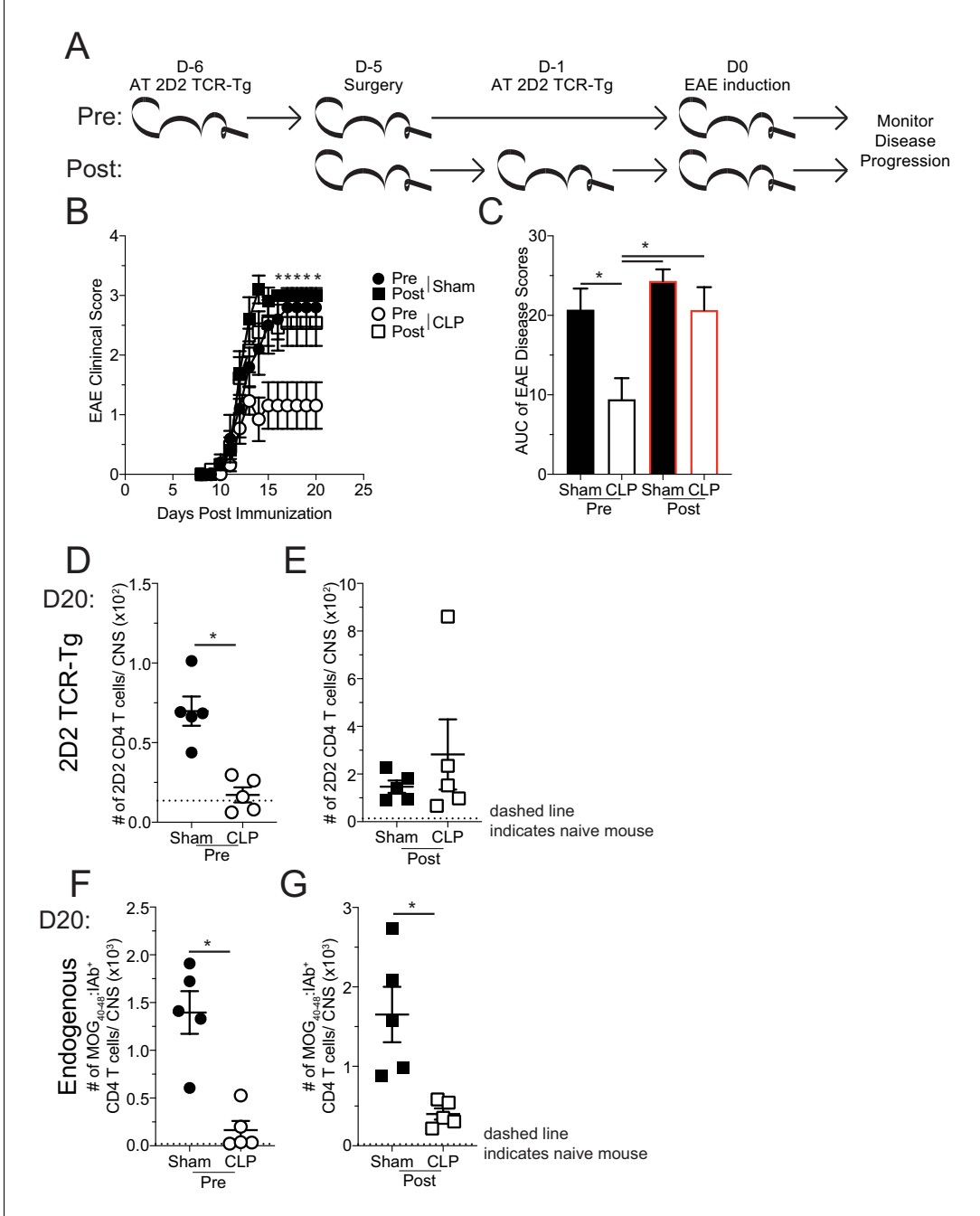

**Figure 8.** Sepsis-induced numerical loss of MOG-specific naive CD4 T cell precursors diminishes EAE disease development. (**A**) Experimental design: Thy1.2 C57Bl/6 mice were separated into Pre- and Post-transfer groups. The Pre-transfer group received $5 \times 10^3$ naive Thy1.1 2D2 TCR-Tg CD4 T cells 1 day before sham or CLP surgery. The Post-transfer group underwent sham or CLP surgery then received $5 \times 10^3$ naive Thy1.1 2D2 TCR-Tg CD4 T cells 4 days later. EAE was induced in both the Pre- and Post-transfer groups 5 days after surgery (Pre: 6 days post 2D2 T cell transfer; Post: 1 day post 2D2 T cell transfer). EAE disease onset and severity were monitored. (**B**) Average EAE disease score following EAE induction for mice that were Pre-transfer sham (black circle), Post-transfer sham (black square), Pre-transfer CLP (white circle), or Post-transfer CLP (white square). (**C**) Area under the curve (AUC) of disease scores in panel B following EAE induction. Number of 2D2 TCR-Tg CD4 T cells in the CNS of sham and CLP (**D**) Pre-transfer and (**E**) Post-transfer groups as well as endogenous MOG-specific CD4 T cells in the CNS of sham and CLP for (**F**) Pre-transfer and (**G**) Post-transfer groups. Data are representative from two independent experiments with 10–14 mice per group for panels **B** and **C**, and five mice per group for panels **D-G**. *p<0.05. Error bars represent the standard error of the mean.

The online version of this article includes the following source data for figure 8:

**Source data 1.** Source data for *Figure 8*.
**Source data 2.** Source data for *Figure 8*.

Pivotal to our findings were the enumeration of MOG-specific CD4 T cell precursors and use of TCR-transgenic 2D2 CD4 T cells. With these tools, we established that sepsis diminishes the number of MOG-specific CD4 T cell precursors but does not alter capacity of the surviving cells to proliferate (either by intrinsic or extrinsic mechanisms) or promote disease if numerically bolstered. This does not, however, suggest that intrinsic and extrinsic factors are non-consequential to CD4 T cells in individuals that survive a septic event. Rather, this is likely reflective of the system of disease induction used. While a large bolus of antigen and adjuvant in the immunization strategy is likely able to overcome the intrinsic and extrinsic defects observed in other systems, this proven experimental model has given us the ability to further interrogate how the sepsis-induced cytokine storm influences the naive CD4 T cell repertoire and development of autoimmunity.

In the context of EAE, our findings suggest that the number of MOG-specific naive CD4 T cell precursors is a critical determining factor in the development of disease. The significance of naive myelin antigen-specific CD4 T cell precursors in the development of MS is highlighted by studies showing that genetic and environmental factors influence the MS incidence and severity through modulation of autoreactive naive T cell precursor frequency. Specifically, allelic differences in HLA-expression may influence the development of MS by differential generation of autoreactive naive CD4 T cells (*Patsopoulos, 2018*). Environmental factors can also influence the precursor frequency either by influencing central tolerance (i.e. vitamin-D-regulated HLA DRB1*1501 expression) or peripheral tolerance (i.e. gut-microbe-regulated Treg frequency and function) (*Freedman et al., 2018*; *Ramagopalan et al., 2009*). As antigen-specific CD4+ T cell responses play a significant role in the pathogenesis of MS, it is plausible that sepsis can modulate the development of MS and/or disease relapses. However, there are no clinical data on the effect of sepsis on the disease development or relapses in patients with MS. Additionally, we observed a numerical recovery of MOG-specific CD4 T cells with time after sepsis in some animals. Yet, the recovered MOG-specific CD4s from CLP hosts may still exhibit inherent differences from MOG-specific CD4s within sham counterparts (e.g. potential reduced avidity of 'recovered' cells). Indeed, the acquisition of an antigen-experienced phenotype by MOG-specific CD4 T cells from CLP mice demonstrates one such difference that may influence the function of these CD4 T cells. Auto-reactive T cells from MS patients have several functional differences, including increased survival capacity, distinct transcriptomic profiles, and increased avidity for autoantigen (*Bieganowska et al., 1997*; *Bielekova et al., 2004*; *Cao et al., 2015*; *Hong et al., 2004*). Therefore, evaluating how sepsis alters the survival capacity, transcriptomic profile, and auto-antigen avidity of cells may further define/elucidate novel underlying mechanisms associated with the development of autoimmune disease.

However, many other prominent questions from the sepsis perspective also remain unanswered. Foremost is the durability of sepsis-induced suppression on the development of autoimmunity and whether the causal impairment remains consistent over that time period. The reduced number of MOG-specific CD4 T cell precursors seems to be a factor at a time proximal to sepsis induction. Additionally, this appeared to extend to the later day 25 timepoint in some mice. A potential explanation for the duality in disease development at day 25 may be the severity of the septic event, though this remains unclear. However, previous work has described how some CD4 T cell precursors eventually recover to pre-sepsis numbers with time, whereas others remain numerically decreased or can even be numerically expanded (*Cabrera-Perez et al., 2015*). In particular, those cells that had numerically expanded had encountered cognate antigen, so the numerical 'recovery' may alternately represent an antigen-specific response due to release of antigen during sepsis-induced cell death. This would be consistent with the increase in CD44 expression by MOG-specific cells. Further, T cells that expand via homeostatic proliferation in lymphopenic states, such as the post-septic environment, can gain an antigen-experienced phenotype and effector function (*Hamilton et al., 2006*). Along with decreased barrier integrity after sepsis (*Honig et al., 2016*), the potential of sepsis promoting autoimmune disease development remains an intriguing alternative to a continuation in the impediment of autoimmunity described here. Yet, given that not all mice develop severe disease this may indicate that at this late timepoint additional and potential novel factors may be at play in the suppression of autoimmune disease development.

Interrogating the impact of sepsis on other autoimmune diseases, in particular those with different effector cells, rather than the CD4 T cells described here, could provide further robust characterization of the sepsis-induced immunoparalysis state. Characterization of how sepsis also influences cells that function to suppress autoimmune disease, such as CD4 $T_{regs}$ which are more resistant to sepsis-induced numerical loss than other CD4 T cell subsets (*Cavassani et al., 2010*; *Monneret et al., 2003*; *Scumpia et al., 2006*; *Sharma et al., 2015*), would also provide an interesting angle for interrogating the immunoparalysis state. In addition, neuro-regulatory CD8 T cells have recently been described to contribute to protection against EAE (*Sinha et al., 2015*). While undescribed in the context of sepsis, the influence of sepsis on these cells may have divergent outcomes for the host. If, similar to CD4 $T_{regs}$, these neuro-regulatory CD8 T cells are more resistant to sepsis-induced lymphopenia this may be an additional mechanism by which autoimmune disease could be suppressed. Alternately, if their function is impaired by sepsis, similar to known impacts on effector CD8 T cells (*Duong et al., 2014*), this could potentiate or exacerbate disease when precursor loss is not the dominant factor. Another factor to consider is that sepsis alters the composition of the host intestinal microbiome (*Alverdy and Krezalek, 2017*; *Krezalek et al., 2016*; *Zaborin et al., 2014*). Environmental factors that the host experiences, such as the microbiome, account for ~70% of MS disease risk (*Kuusisto et al., 2008*), and there is growing evidence that the host's intestinal microbiome actively influences MS disease (*Chen et al., 2016*; *Freedman et al., 2018*). Thus, interrogation into how the sepsis-induced changes in intestinal microbiota correspond with microbiomes associated with MS may elicit further insight into the interaction between these disease states. Finally, the mechanism/timing of the initiation of other autoimmune model systems should also be a strong consideration as it could inform on different aspects of the immunoparalysis state and generate distinctions between newly developed and established autoimmunity.

The novel characterization of how infection, in the form of a septic insult, can dramatically influence the development of autoimmunity, presented here, reframes the complexity of the immunoparalysis state. By interrogating the impact of sepsis on inflammatory states beyond infection/cancer, additional mechanisms of sepsis-induced impairment may become apparent. Further, by understanding how sepsis influences these diseases, insights into the mechanisms that underlie their pathologies might also be illuminated. Thus, a final pertinent direction for future study would be to understand the interaction between sepsis and autoimmunity in patient cohorts. While it appears that patients with autoimmunity, and MS in particular, have a higher incidence of sepsis (*Capkun et al., 2015*; *Nelson et al., 2015*), the reverse scenario is still unexamined at this time. The added burden of sepsis in patients with autoimmunity is likely associated with the inflammatory status of the host, similar to the increased susceptibility of 'dirty' mice to a septic event (*Huggins et al., 2019*). However, similar to our results, the immunoparalysis state may, in fact, benefit those patients who survive the cytokine storm by reducing the function of the pathogenic autoreactive cells. Such an outcome would be further instructive in understanding the interplay between autoimmunity and the infectious events (e.g. sepsis).

# Materials and methods

**Key resources table**

| Reagent type (species) or resource | Designation | Source or reference | Identifiers | Additional information |
|---|---|---|---|---|
| Strain, strain background (*Mus musculus*) | C57BL6/J | Jackson Laboratory | Stock No: 000664 (RRID:IMSR_ JAX:000664) | |
| Strain, strain background (*Mus musculus*) | B6.PL(84NS)/Cy | Jackson Laboratory | Stock No: 000983 (RRID:IMSR_ JAX:000406) | C57BL6/J Thy1.1 |
| Strain, strain background (*Mus musculus*) | C57BL/6-Tg(Tcra2D2, Tcrb2D2)1Kuch/J | Jackson Laboratory | Stock No: 006912 (RRID:IMSR_ JAX:006912) | |

*Continued on next page*

Continued

| Reagent type (species) or resource | Designation | Source or reference | Identifiers | Additional information |
|---|---|---|---|---|
| Strain, strain background (*Mus musculus*) | Thy1.1/1.1-C57BL/6-Tg(Tcra2D2, Tcrb2D2)1Kuch/J | This paper | Thy1.1/1.1 2D2 | Can be acquired through lab contact or breeding of above commercially available strains |
| Strain, strain background (*Mus musculus*) | Thy1.1/1.2-C57BL/6-Tg(Tcra2D2, Tcrb2D2)1Kuch/J | This paper | Thy1.1/1.2 2D2 | Can be acquired through lab contact or breeding of above commercially available strains |
| Peptide, recombinant protein | MOG$_{35-55}$ | GenScript | SC1208 | |
| Other | CFA containing *M. tuberculosis* H37Ra | Difco | DF3114-33-8 | |
| Peptide, recombinant protein | Pertussis toxin from Bordetella pertussis | Sigma-Aldrich | P7208 | |
| Antibody | CD4 (Rat monoclonal) | Biolegend | GK1.5 (AB_312689) | FACs (1:400) |
| Antibody | CD11a (Rat monoclonal) | Biolegend | M17/4 (AB_312776) | FACs (1:300) |
| Antibody | IFNγ (Rat monoclonal) | eBioscience | XMG1.2 (AB_465410) | FACs (1:100) |
| Antibody | IL-17A (Rat monoclonal) | eBioscience | eBio17B7 (AB_906240) | FACs (1:100) |
| Antibody | TNFα (Rat monoclonal) | eBioscience | MP6-XT22 (AB_465416) | FACs (1:100) |
| Antibody | CD8a (Rat monoclonal) | Biolegend | 5H10-1 (AB_312762) | FACs (1:400) |
| Antibody | Ki67 (Mouse monoclonal) | BD Pharmingen | B56 (AB_2858243) | FACs (1:100) |
| Antibody | Thy1.1 (Mouse monoclonal) | eBioscience | HIS51 (AB_1257173) | FACs (1:1000) |
| Antibody | Thy1.2 (Rat monoclonal) | eBioscience | 53–2.1 (AB_467378) | FACs (1:1000) |
| Antibody | CD44 (Rat monoclonal) | eBioscience | IM7 (AB_469715) | FACs (1:200) |
| Antibody | CD45 (Mouse monoclonal) | eBioscience | 104 (AB_469724) | FACs (1:100) |
| Antibody | F4/80 (Rat monoclonal) | Biolegend | BM8 (AB_893499) | FACs (1:100) |
| Antibody | CD11b (Rat monoclonal) | eBioscience | M1/70 (AB_468883) | FACs (1:200) |
| Antibody | IA-b (Rat monoclonal) | eBioscience | M5/114.15.2 (AB_529608) | FACs (1:100) |
| Antibody | CD3e (Armenian Hamster monoclonal) | eBioscience | 145–2 C11 (AB_467048) | FACs (1:100) |
| Antibody | CD19 (Mouse monoclonal) | eBioscience | MB19-1 (AB_467145) | FACs (1:100) |

*Continued*

| Reagent type (species) or resource | Designation | Source or reference | Identifiers | Additional information |
|---|---|---|---|---|
| Antibody | FAS (Mouse monoclonal) | Biolegend | SA367H8 (AB_2629777) | FACs (1:100) |
| Antibody | FASL (Armenian Hamster monoclonal) | Biolegend | MFL3 (AB_313276) | FACs (1:100) |
| Antibody | TRAIL (Rat monoclonal) | Biolegend | N2B2 (AB_345271) | FACs (1:100) |
| Antibody | RORγT (Rat monoclonal) | eBioscience | AFKJS-9 (AB_1834470) | FACs (1:100) |
| Antibody | Tbet (Mouse monoclonal) | eBioscience | eBio4b10 (AB_763636) | FACs (1:100) |
| Antibody | FoxP3 (Rat monoclonal) | Invitrogen | FJK-16S (AB_467576) | FACs (1:100) |
| Antibody | CD11c (Armenian Hamster monoclonal) | Biolegend | N418 (AB_313772) | FACs (1:100) |
| Antibody | B220 (Rat monoclonal) | Biolegend | RA3-6B2 (AB_312989) | FACs (1:100) |
| Peptide, recombinant protein | MOG$_{40-48}$ I-Ab linked (*Drosophila melanogaster* S2 cells) | NIH tetramer core | | FACs (1:100) |
| Commercial assay or kit | Foxp3 / Transcription Factor Staining Buffer Set | Invitrogen | 00-5523-00 | |
| Commercial assay or kit | Vybrant FAM Caspase-3 and −7 assay kit | Thermo-Fischer | V35118 | |
| Commercial assay or kit | BioRad Bio-plex Pro Mouse Cytokine 23-plex | Biorad | M60009RDPD | |
| Software, algorithm | GraphPad Prism | GraphPad Prism 8 | Version 8.4.2 (464) (RRID:SCR_002798) | |

## Mice

Inbred C57Bl/6 (B6; Thy1.2/1.2) were purchased from the National Cancer Institute (Frederick, MD) and maintained in the animal facilities at the University of Iowa at the appropriate biosafety level. 2D2 (C57BL/6-Tg(Tcra2D2,Tcrb2D2)1Kuch/J) mice on the C57BL6/J background were purchased from Jackson Laboratories and bred with C57BL6/J (Thy1.1/1.1), purchased from the National Cancer Institute (Frederick, MD), to generate heterozygotes for 2D2 and Thy1.1. $F_1$ mice were bred together to generate 2D2 Thy 1.1/1.1 and 2D2 Thy1.1/1.2 mice. Expression of the 2D2 TCR and Thy1.1/1.1 were confirmed by flow cytometric staining.

## Cecal ligation and puncture (CLP) model of sepsis induction

Mice were anesthetized with ketamine/xylazine (University of Iowa, Office of Animal Resources), the abdomen was shaved and disinfected with Betadine (Purdue Products), and a midline incision was made. The distal third of the cecum was ligated with Perma-Hand Silk (Ethicon), punctured once

using a 25-gauge needle, and a small amount of fecal matter extruded. The cecum was returned to abdomen, the peritoneum was closed with 641G Perma-Hand Silk (Ethicon), and skin sealed using surgical Vetbond (3M). Following surgery, 1 mL PBS was administered s.c. to provide post-surgery fluid resuscitation. Lidocaine was administered at the incision site, and flunixin meglumine (Phoenix) was administered for postoperative analgesia. This procedure created a septic state characterized by loss of appetite and body weight, ruffled hair, shivering, diarrhea, and/or periorbital exudates with 0–10% mortality rate. Sham mice underwent identical surgery excluding cecal ligation and puncture.

### EAE disease induction and evaluation

EAE was induced and evaluated as shown previously (*Mangalam et al., 2009*). Briefly, mice were immunized s.c. on day 0 on the left and right flank with 100 µg of $MOG_{35-55}$ emulsified in Complete Freund's Adjuvant followed by 80 ng of pertussis toxin (PTX) i.p. on days 0 and 2. Disease severity was scored as follows: 0, no clinical symptoms; 1, loss of tail tonicity; 2, hind limb weakness; 3, hind limb paralysis; 4, fore limb weakness; 5, moribund or death.

### Cell isolation

Single-cell suspensions from lymph nodes and spleens were generated after mashing tissue through 70-µm cell strainer without enzymatic digestion. To isolate CNS leukocytes, mice were anesthetized with $CO_2$ and quickly perfused through the left ventricle with cold PBS. Brains were removed from the skull and spinal cords were flushed through the vertebral canal with cold RPMI media. To isolate immune cells from the CNS, brain and spinal cords were combined, homogenized, and isolated by Percoll gradient centrifugation. Following centrifugation, CNS leukocytes were collected from the interface, washed, and prepared appropriately for further use.

### Histology

Mice were euthanized using $CO_2$ and intravascularly perfused using a gravity fed system with 10% neutral buffered (10% NBF) formalin via intracardiac puncture. Spinal cords were then emersion fixed in 10% NBF for another 24–48 hr. Spinal cords were left in situ, demineralized with 14% EDTA for ~4 days and then embedded in paraffin and routine processed. Sections (4 µm thick) were stained with hemotoxylin and eosin (H and E) and Luxol fast blue (LFB) and analyzed by a board-certified veterinary pathologist. Spinal cord sections were scored for cord pathology and meningeal inflammation. The meningeal score was a 0 to 4 scale where 0 = no pathology; 1 = rare, scattered, mild meningeal inflammatory cell infiltrates; 2 = mild, multifocal and obvious meningeal inflammatory cell infiltrates; 3 = multifocal to coalescing meningeal inflammatory cell infiltrates; 4 = marked, diffuse, thick bands of meningeal inflammatory cell infiltrates. The spinal cord score identifies how much of the cord at that level was affected and was also a 0 to 4 scale where 0 = no pathology; 1 = 1–25% of the spinal cord is affected with pathology consistent with EAE; 2 = 30–50% of the spinal cord is affected with pathology consistent with EAE; 3 = 60–90% of the spinal cord is affected with pathology consistent with EAE; 4 = >90% of the spinal cord is affected with pathology consistent with EAE.

### Flow cytometry and intracellular protein detection

Flow cytometry data were acquired on a FACSCanto (BD Biosciences, San Diego, CA) and analyzed with FlowJo software (Tree Star, Ashland, OR). To determine expression of cell surface proteins, mAb were incubated at 4°C for 20–30 min and cells were fixed using Cytofix/Cytoperm Solution (BD Biosciences) and, in some instances followed by mAb incubation to detect intracellular proteins. The following mAb clones were used: CD4 (GK1.5, Biolegend), CD11a (M17/4, Biolegend), IFNγ (XMG1.2; eBioscience), IL-17A (eBio17B7, eBioscience), TNFα (MP6-XT22, eBioscience), CD8a (5H10-1, Biolegend), Ki67 (B56, BD Pharmingen), Thy1.1 (HIS51, eBioscience), Thy1.2 (53–2.1, eBioscience), CD44 (IM7, eBioscience), CD45 (104, eBioscience), F4/80 (BM8, Biolegend), CD11b (M1/70, eBioscience), IA-b (M5/114.15.2, eBioscience), CD3e (145–2 C11, eBioscience), CD19 (MB19-1, eBioscience), FAS (SA367H8, Biolegend), FASL (MFL3, Biolegend), TRAIL (N2B2, Biolegend), RORγT (AFKJS-9, eBioscience), Tbet (eBio4B10, eBioscience), FoxP3 (FJK-16S, Invitrogen), and Dump [CD11b (M1/70), CD11c (N418), B220 (RA3-6B2), F4/80 (BM8), BioLegend].

Enrichment and detection of endogenous MOG-specific CD4 T cells: Biotinylated I-A$^b$ molecules containing the MOG$_{40-48}$ epitope covalently linked to the I-A$^b$ β-chain were produced in *Drosophila melanogaster* S2 cell along with the I-A$^b$ β-chain (*Moon et al., 2007*). The monomers were purified, and then made into tetramers with streptavidin-phycoerythrin (SA-PE; Prozyme). To enrich for Ag-specific CD4 T cells, tetramers (10 nM final concentration) were then added to single-cell suspensions in 300 μl tetramer staining buffer (PBS containing 5% FBS, 2 mM EDTA, 1:50 normal mouse serum, and 1:100 anti-CD16/32 mAb). The cells were incubated in the dark at room temperature for 1 hr, followed by a wash in 10 ml ice cold FACS Buffer. The tetramer-stained cells were then resuspended in 300 μl FACS Buffer, mixed with 25 μl of anti-PE mAb-conjugated magnetic microbeads (StemCell Technologies), and incubated in the dark on ice for 30 min. The cells were washed, resuspended in 3 ml cold FACS Buffer, and passed through an EasySep Magnet (StemCell Technologies) to yield the enriched tetramer positive population. The resulting enriched fractions were stained with a cocktail of fluorochrome-labeled mAb: Thy1.2, CD4, CD8, CD44, 'dump' (CD11b, CD11c, B220, F4/80), and tetramer. Cell numbers for each sample were determined using AccuCheck Counting Beads (Invitrogen). Samples were then analyzed using a Fortessa flow cytometer (BD) and FlowJo software.

Intracellular cytokine staining: For direct ex vivo staining, cells were incubated for one additional hour in the presence of Brefeldin A (BFA) before surface and intracellular cytokine staining.

Ki67 staining: Following surface staining cells were fixed overnight with FoxP3 fixation/permeabilization buffer then stained with Ki67.

Propidium Iodide and active Caspase 3/7 staining: Vybrant FAM Caspase-3 and −7 assay kit (Thermo-Fischer) was used to identify apoptotic cells via expression of active caspase3/7 and propidium iodide according to the manufacturer's instructions. Briefly cells were incubated with FLICA reagent for 30 min at 37°C followed by surface staining with antibodies as well as propidium iodide at 4°C for 20 min. Cells were immediately analyzed after staining without fixation by flow cytometry.

## Adoptive transfer (AT) of 2D2 cells

For 2D2 transfers 200 μl of blood was collected from Thy1.1/1.1 or Thy1.1/1.2 2D2 mice in heparin-coated capillary tubes or spleens were harvested and homogenized. Red blood cells were lysed and the frequency of naive CD4 T cells was determined by flow cytometric analysis of a portion of the samples. Remaining cells were then enumerated and then adjusted to transfer $5 \times 10^3$ naive CD4 T cells per mouse prior to immunization or $5 \times 10^6$ naive CD4 T cells per mouse to assess cell survival in non-immunized mice. Cells were transferred via retroorbital injection.

## Multiplex cytokine analysis

Multiplex cytokine analysis was performed via BioRad Bio-plex Pro Mouse Cytokine 23-plex according to the manufacturer's instructions for plasma cytokine analysis. Multiplex was analyzed on BioRad Bio-Plex (Luminex 200) analyzer in the university of Iowa Flow Cytometry core facility.

## Statistical analysis

Unless stated otherwise data were analyzed using Prism eight software (GraphPad) using two-tailed Student t-test (for two individual groups, if variance was unequal variance then Mann-Whitney U test), one-way ANOVA with Bonferroni post-hoc test (for >2 individual groups, if variance was unequal variance then Kruskal-Wallis with Dunn's post-hoc test was used), two-way ANOVA (for multiparametric analysis of two or more individual groups, pairing was used for samples that came from the same animal) with a confidence interval of >95% to determine significance (*p<0.05). Log-rank (Mantel-Cox) curve comparisons was used to determine significant difference in time to disease EAE disease onset (*p<0.05). Data are presented as standard error of the mean.

## Acknowledgements

We thank members of our laboratories and the lab of Dr. Karandikar for technical assistance and helpful discussions. In particular, we wish to thank Nicole Cady for animal care and maintenance of transgenic mice. Further, we wish to thank Adam Geoken, Thomas Businga, and the comparative pathology laboratory for tissue processing and histochemical staining as well as the University of Iowa Flow Cytometry Core Facility staff.

## Additional information

### Funding

| Funder | Grant reference number | Author |
|---|---|---|
| National Institute of Allergy and Infectious Diseases | AI114543 | Vladimir P Badovinac |
| National Institute of Allergy and Infectious Diseases | AI147064 | Vladimir P Badovinac |
| National Institute of General Medical Sciences | GM113961 | Vladimir P Badovinac |
| National Institute of General Medical Sciences | GM134880 | Vladimir P Badovinac |
| National Institute of General Medical Sciences | GM115462 | Thomas S Griffith |
| National Institute of Allergy and Infectious Diseases | AI137075 | Ashutosh K Mangalam |
| U.S. Department of Veterans Affairs | I01BX001324 | Thomas S Griffith |
| National Institute of Environmental Health Sciences | P30 ES005605 | Katherine N Gibson-Corley Ashutosh K Mangalam |
| National Institute of Diabetes and Digestive and Kidney Diseases | 5P30DK054759 | Katherine N Gibson-Corley |
| National Institute of Allergy and Infectious Diseases | T32AI007511 | Isaac J jensen |
| National Institute of Allergy and Infectious Diseases | T32AI007485 | Isaac J jensen Samantha N Jensen |
| National Institute of Allergy and Infectious Diseases | AI137075-S1 | Samantha N Jensen |
| National Cancer Institute | T32CA009138 | Frances V Sjaastad |
| National Institute of Allergy and Infectious Diseases | T32AI007313 | Frances V Sjaastad |
| National Institute of General Medical Sciences | 1R35134880 | Vladimir P Badovinac |

The funders had no role in study design, data collection and interpretation, or the decision to submit the work for publication.

### Author contributions

Isaac J Jensen, Samantha N Jensen, Conceptualization, Data curation, Formal analysis, Investigation, Writing - original draft, Writing - review and editing; Frances V Sjaastad, Formal analysis, Investigation; Katherine N Gibson-Corley, Formal analysis; Thamothrampillai Dileepan, Resources; Thomas S Griffith, Conceptualization, Writing - review and editing; Ashutosh K Mangalam, Conceptualization, Resources, Supervision, Funding acquisition, Writing - review and editing; Vladimir P Badovinac, Conceptualization, Resources, Supervision, Funding acquisition, Visualization, Writing - review and editing

### Author ORCIDs

Isaac J Jensen https://orcid.org/0000-0002-3107-3961
Samantha N Jensen https://orcid.org/0000-0002-3001-5217
Thomas S Griffith http://orcid.org/0000-0002-7205-9859
Vladimir P Badovinac https://orcid.org/0000-0003-3180-2439

## Ethics

Animal experimentation: Experimental procedures using mice were approved by University of Iowa Animal Care and Use Committee under ACURF protocol number 9101915. The experiments performed followed Office of Laboratory Animal Welfare guidlines and PHS policy on Humane Care and Use of Laboratory Animals. Euthansia was performed by cervical dislocation or carbon dioxide asphyxiation.

## Decision letter and Author response

Decision letter https://doi.org/10.7554/eLife.55800.sa1
Author response https://doi.org/10.7554/eLife.55800.sa2

# Additional files

## Supplementary files

• Transparent reporting form

## Data availability

All data generated or analysed during this study are included in the manuscript and supporting files. Source data files have been provided.

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
