## [Decision Letter]

**Acceptance summary:**

It has been established that sepsis exacerbates various disease states, including cancer, by causing immunoparalysis. In this manuscript, using well established mouse model of autoimmunity, the authors depart from this view by proposing that sepsis may actually benefit experimental autoimmune condition owing to the inflammatory cytokines that may affect antigenic response of cells types that are involved in autoimmunity.

**Decision letter after peer review:**

Thank you for submitting your article "Sepsis impedes EAE disease development by diminishing autoantigen-specific naïve CD4 T cells" for consideration by *eLife*. Your article has been reviewed by two peer reviewers, and the evaluation has been overseen by a Reviewing Editor and Satyajit Rath as the Senior Editor. The following individuals involved in review of your submission have agreed to reveal their identity: Pawn Kuman (Reviewer #1); Sofia Casares (Reviewer #2).

The reviewers have discussed the reviews with one another and the Reviewing Editor has drafted this decision to help you prepare a revised submission. In recognition of the fact that revisions may take longer than the two months we typically allow, until the research enterprise restarts in full, we will give authors as much time as they need to submit revised manuscripts. With the authors agreement, we will post this manuscript to bioRxiv along with the reviews and a formal declaration that the manuscript is "in revision at *eLife*".

Summary:

The study on sepsis-induced immunosuppression of myelin oligodendrocyte (MOG)-specific naïve CD4 T cells interesting. The manuscript is well organized and well-written. For the most part, the results support the authors' conclusions that experimentally induced sepsis leads to immunosuppression, with the expectation of ameliorating EAE in a mouse model.

Essential revisions:

1) Address experimentally if the number of MOG-specific CD4 T cell precursors persist for longer time in CLP mice, under conditions where EAE is initiated on day 20 or 30 post-surgery on CLP survivor.

2) Apart from the results showing the reduction of CD4 T cells and effector cytokines on day 15 post EAE, please show effector cytokines/chemokines (GM-CSF, IL-12, IL-23, IFNγ, G-CSF, IL-17A, IL-10) levels in the serum of day 0 and day 15 EAE mice.

Is there a difference in infiltrating inflammatory myeloid cell number in the CNS of EAE induced mice?

3) Include Th1, Th17 and Treg frequency and absolute number in pre- and post-transfer groups.

4) Consider a CLP experiment in 2D2 transgenic mice and transfer activated immune cells to a new host for EAE experiment.

5) Show FAS, TRAIL, FasL expression and apoptotic cells (Annexin and 7AAD) of the transferred group (Figure 6).

6) Include histopathology results (Figure 1 E, F, G) for non-surgery mice. These data should be added to determine whether differences exist for spinal cord and meningeal disease between non-surgery mice and the other two groups. Again the effect of anesthesia.

7) Provide an in-depth discussion on the relationship between your current findings in the context of understanding MS pathogenesis or autoimmunity.

8) Present data or at least discuss the potential effects of anesthesia (sham vs. non-surgery groups) on T cell responses. Results on no-surgery groups would be critical to assess the effect of anesthesia on the MOG reactive and non-reactive CD4 T cells.

[Editors' note: further revisions were suggested prior to acceptance, as described below.]

Thank you for resubmitting your work entitled "Sepsis impedes EAE disease development by diminishing autoantigen-specific naïve CD4 T cells" for further consideration by *eLife*. Your revised article has been reviewed by three peer reviewers, and the evaluation has been overseen by a Reviewing Editor and Satyajit Rath as the Senior Editor.

The manuscript has been improved but there are some remaining issues that need to be addressed before acceptance, as outlined below:

Upon review of your resubmitted manuscript, the reviewers considered the study to be of interest, the experiments solidly performed, and the manuscript well written. However, major issues remain with your conclusion or the major claim, namely that the lower number of naive T cells following CLP leads to reduced EAE. The reviewers thought that the interpretation of the data is not completely correct as the link between the reduced naïve T cell numbers and reduced disease outcome has not been proven and thus the discussion remains too speculative. In the absence of conclusive evidence, other scenarios, which were not considered, should also be discussed. For example, it is likely that following CLP, a strong pro-inflammatory state arose the mice due to a high production of IL-1, IL-6 and TNF. This could have resulted in a reduced level of MOG-specific T cells but more importantly, it could have led to a general tolerogenic state that temporarily prevented an efficient immune response from occurring. The authors should also consider the possibility that alteration in Treg frequency/function in the post-septic mice, or a difference amongst other cell types such as APCs, or other such factors.

Another likely interpretation is that the CLP and the massive depletion that followed and led to lymphopenia also caused a general state of tolerance. An expansion of Treg cells that can dominantly inhibit priming of MOG-specific T cells could also be considered. Here, the authors show that the massive T cell killing after CLP partially spared Treg cells as their proportion was increased compared to the effector T cells. Another possibility is that there was an expansion of tolerogenic dendritic cells or the efficiency of the DCs was harmed by CLP.

Therefore, it has been suggested that the Abstract, Discussion, as well as the title of your manuscript be revised. The changes should reflect the fact that you have not demonstrated that the reason for the diminished EAE is indeed the reduction of naive T cells (unless you can, in fact, provide such data, ruling out alternative interpretations).

There are other specific questions that arose given the interpretation of the current data. The authors show the group of mice that received 2D2 T cells prior to CLP showed a similar disease reduction as mice that are only immunized without receiving any additional T cells (Figure 1). Even if the number of originally transferred 2D2s (5x10^6^) was reduced after CLP compared to the post-CLP transfer (Figure 6—figure supplement 2), one would still expect to have a significantly increased number of MOG-specific T cells in the system compared to mice that did not receive any T cells. This then casts doubt on the fact that the reduced number of naïve T cells in post-septic mice is the reason for the EAE amelioration. The authors alternatively suggest a potential priming deficit as an explanation for their phenotype, however, they do not follow up on this. They indeed see a reduced frequency of MOG-specific T cells in the draining lymph nodes at day 7 post immunization (Figure 4), but do not explain a potential mechanism.

1) Figure 1A the very first finding to start off the whole story, shows only up to day 13. This is way too short for evaluating EAE and certainly at the suboptimal quality of its experimental setup. Because of it, EAE with CLP may be just simply delayed, and the authors cannot claim the EAE "ablated" and "Reduction in disease severity."

2) Why was CD11a used to gate on T cells? Normally one would gate on a T cell marker such as CD90 or just CD4.

3) In Figure 3, please show a dot plot for each case (Sham and CLP) for IFNγ versus IL-17A and one of the two versus TNFa, or even better, GM-CSF versus TNFa. FMO can be shown in the Supplementary figure showing the data as histograms is not informative.

4) What is the gate used in Figure 6E?

---

## [Author Response]

Essential revisions:1) Address experimentally if the number of MOG-specific CD4 T cell precursors persist for longer time in CLP mice, under conditions where EAE is initiated on day 20 or 30 post-surgery on CLP survivor.

This question is important given that sepsis can lead to long-lasting effects on the host’s immune system. We have added new data demonstrating that at 25 days post-sepsis mice still developed less severe disease than sham controls, although the EAE was more pronounced when compared to day 5 CLP counterparts (new Figure 1B-D). This was potentially due to significant variability in EAE development in individual mice – some mice developed severe disease (similar to the control, Sham mice) while others did not develop robust disease (similar to day 5 CLP mice).

This new data suggested that numerical recovery of naïve MOG-specific CD4 T cell precursors may occur within this 25 day timeframe, at least in some of the mice. Indeed, when we subsequently quantitated the number of naïve precursors at day 25 post-CLP induction in separate cohorts of mice, we observed that numerical recovery of naïve MOG-specific CD4 T cell pool occurred in some, but not all of the mice (new Figure 5D) leading to statistically insignificant changes in the number of naïve CD4 T cell precursors in both groups of mice at this time point after surgery. However, it is tempting to speculate that this variability in numerical recovery is in agreement with observed variability in disease development (new Figure 1B-D), suggesting that numerical impairment may still control the susceptibility to the disease development and progression. Given that it is not possible to assess the CD4 T cell precursor frequency and the capacity to develop disease in the same mouse this remains as an intriguing correlation.

Interestingly, we also observed that the naïve MOG-specific CD4 T cells present in the CLP hosts are phenotypically distinct and resemble “antigen-experienced” CD44^hi^ cells (new Figure 5E,F). It is unclear whether these cells become antigen-experienced through recognition of cognate antigen released during septic insult and/or due to the well-described phenomenon of homeostatic proliferation in sepsis-induced lymphopenic environment. Thus, changes in the function of MOG-specific CD4 T cells present in sepsis survivors might also contribute to the observed differences in EAE induction, an intriguing notion to be pursued in the future.

Text that include discussion of these data was modified accordingly throughout the revised version of the manuscript.

2) Apart from the results showing the reduction of CD4 T cells and effector cytokines on day 15 post EAE, please show effector cytokines/chemokines (GM-CSF, IL-12, IL-23, IFNγ, G-CSF, IL-17A, IL-10) levels in the serum of day 0 and day 15 EAE mice.Is there a difference in infiltrating inflammatory myeloid cell number in the CNS of EAE induced mice?

We performed the requested cytokine analysis in the serum and demonstrated sham mice had a distinct signature of cytokine expression following disease (day 15 after EAE induction compared to day 0) which was not mirrored within CLP groups of mice (new Figure 1—figure supplement 1A).

Of note: While many cytokines/chemokines did not exhibit significant changes between days 0 and 15 after EAE induction this is in agreement with previous reports describing that the majority of inflammation is localized inside CNS (Pierson et al., 2012 Immunol Rev 248(1); Pareek et al., 2011 Sci Rep 1:201). Consequently, we analyzed the inflammatory myeloid cells in the CNS of sham and CLP hosts. The frequency and number of both microglia and infiltrating monocytes and macrophages were notably diminished in CLP hosts (new Figure 1—figure supplement 2A-E). This change in frequency/number of myeloid cells was further compounded by a decrease in MHC II expression (both a marker of myeloid cell activation and means of antigen presentation to CD4 T cells) on both of these cell populations (new Figure 1—figure supplement 2F-I).

Cumulatively, these new data further demonstrate that CLP hosts are severely impaired in their capacity to initiate disease in the CNS. This impairment is in agreement with both the ascribed role of encephalitogenic CD4 T cells as critical initiators of EAE and corresponds to our finding of a reduction in CNS-infiltrating, MOG-specific CD4 T cells.

3) Include Th1, Th17 and Treg frequency and absolute number in pre- and post-transfer groups.

New experiments that included pre- and post- transfer groups were performed to address this question. Importantly, we observed a significant numerical loss of all effector populations listed (Th1-Tbet, Th17-RORgT, and Treg-FoxP3) in the pre-transfer group and that decline in numbers is significantly diminished in the post-transfer group, indicating recovery of these CD4 T cell subsets (new Figure 6—figure supplement 2). As an important control, a similar decline in endogenous bulk CD4 T cell populations was observed in pre- and post- transfer groups. These data demonstrate that by overcoming the numerical loss of the MOG-specific CD4 T cells the requisite effector populations needed for EAE disease induction can be generated in vivo.

4) Consider a CLP experiment in 2D2 transgenic mice and transfer activated immune cells to a new host for EAE experiment.

The reviewer’s suggestion for the experiment is insightful and seeks to address intrinsic changes in encephalitogenic naïve MOG-specific CD4 T cells that potentially control the observed lack of EAE induction in CLP hosts. In direct response to this suggestion, we performed sham or CLP surgery on Thy1-disparate 2D2 TCR-Tg CD4 T cell mice and transferred the same number of 2D2 cells (1:1 mix) into Thy1.2 recipients prior to EAE induction (new Figure 7A). The survival of Sham or CLP 2D2 CD4 T cells was indistinguishable as well as their capacity to expand in numbers in draining lymph nodes upon EAE induction (new Figure 7B,C).

Thus, these data suggest sepsis does not lead to measurable cell-intrinsic deficiencies of naive CD4 T cell precursors to respond to auto-antigen and further supports the concept that sepsis-imposed numerical loss of antigen-specific naïve CD4 T cell precursors control the disease development.

5) Show FAS, TRAIL, FasL expression and apoptotic cells (Annexin and 7AAD) of the transferred group (Figure 6).

The reviewers’ question seeks to address whether the septic environment predisposes encephalitogenic CD4 T cells to apoptosis, thereby limiting their expansion and effector potential. Thus, we evaluated the frequency of apoptotic cells (as assessed by active Caspase 3/7 and propidium iodide staining) in the draining lymph nodes in Sham and CLP mice and found no difference in the frequency of apoptotic cells (new Figure 6E). Further, we evaluated the expression the extrinsic apoptosis-related proteins TRAIL, Fas, and FasL, and noted no difference in the expression of these receptors on CD4 T cells in any of the analyzed groups (new Figure 6—figure supplement 1). These new data, thus, further support our conclusion that the numerical loss of the CD4 T cell precursors in sepsis survivors control diminished accumulation of MOG-specific effector CD4 T cells.

6) Include histopathology results (Figure 1 E, F, G) for non-surgery mice. These data should be added to determine whether differences exist for spinal cord and meningeal disease between non-surgery mice and the other two groups. Again the effect of anesthesia.

We recognize this represents an additional control for the histopathology; however, we had not performed histopathology on non-surgery mice for the following reasons: a) Constraints imposed by the coronavirus pandemic on the research enterprise at the University of Iowa prevented us from repeating these experiments to include naïve non-surgery mice as controls (specifically, we were faced with the prolonged closure of the critical Core Research Facility needed for histopathology analyses); and b) Sham surgery represents the most relevant control for CLP procedure – please see our extended discussions related to utility of Sham controls under point #8 and in the revised version of the manuscript.

7) Provide an in-depth discussion on the relationship between your current findings in the context of understanding MS pathogenesis or autoimmunity.

We appreciate the reviewers request to broaden the applicability of our research and have incorporated this Discussion.

8) Present data or at least discuss the potential effects of anesthesia (sham vs. non-surgery groups) on T cell responses. Results on no-surgery groups would be critical to assess the effect of anesthesia on the MOG reactive and non-reactive CD4 T cells.

We recognize that numerous factors associated with surgery may influence subsequent disease outcome. This is why, rather than using non-surgical controls, we utilize Sham surgical controls for the majority of experiments. In particular, the reviewer requests discussion of the influence of anesthesia on T cell responses. In our surgical procedures we use ketamine as an anesthesia and indeed the effect of ketamine on T cell response has been evaluated, including in the context of EAE (Lee et al., 2017). In this study the authors found that increasing concentrations of ketamine, administered in vitro, reduced the capacity of autoantigen-specific CD4 T cells to produce effector cytokines and induce disease. However, this in vitro exposure occurred over the course of several days and does not account for the metabolism of ketamine and excretion through the urinary system (Chang and Glazko 1974 Int Anesthesiol Clin 12(2)).

Since it is our belief that it is important to control for the potential effects of anesthesia and other aspects of surgery, Sham surgery (with the same surgical procedures as CLP, including anesthesia, though performed without cecum ligation and needle puncture) is the most appropriate and preferred control for these experiments.

[Editors' note: further revisions were suggested prior to acceptance, as described below.]

The reviewers thought that the interpretation of the data is not completely correct as the link between the reduced naïve T cell numbers and reduced disease outcome has not been proven and thus the discussion remains too speculative. In the absence of conclusive evidence, other scenarios, which were not considered, should also be discussed. For example, it is likely that following CLP, a strong pro-inflammatory state arose the mice due to a high production of IL-1, IL-6 and TNF. This could have resulted in a reduced level of MOG-specific T cells but more importantly, it could have led to a general tolerogenic state that temporarily prevented an efficient immune response from occurring. The authors should also consider the possibility that alteration in Treg frequency/function in the post-septic mice, or a difference amongst other cell types such as APCs, or other such factors.Another likely interpretation is that the CLP and the massive depletion that followed and led to lymphopenia also caused a general state of tolerance. An expansion of Treg cells that can dominantly inhibit priming of MOG-specific T cells could also be considered. Here, the authors show that the massive T cell killing after CLP partially spared Treg cells as their proportion was increased compared to the effector T cells. Another possibility is that there was an expansion of tolerogenic dendritic cells or the efficiency of the DCs was harmed by CLP.

The reviewer’s above commentary broadly reflects the multitude of events that contribute to the sepsis-induced immunoparalysis state. Indeed, all of these events likely contribute to some degree. These potential CD4 extrinsic influences on disease were discussed in the Introduction, Results and Discussion though additional commentary has been included to further emphasize this point.

Moreover, we experimentally addressed whether the CD4 T cell extrinsic factors may be influencing the development of disease. When we transferred naïve 2D2 CD4 T cells (that did not undergo a septic insult) into mice that either underwent Sham or CLP we observed no difference in the capacity of these 2D2 CD4 T cells to survive, respond to antigen, or promote disease (see post-transfer groups in Figures 6 and 8). If extrinsic factors were restricting the priming of CD4 T cells in the lymph node or preventing induction of disease, then these differences should be observed when comparing post-transfer 2D2 CD4 T cells in Figures 6 and 8 (discussion on reviewer concern about cell transfers is below). We are not suggesting the loss of naïve CD4 T cells is the only mechanism of impairment in the post-septic environment. Rather, our data supports the conclusion that the number of naïve autoantigen-specific CD4 T cells is a critical factor in the development of disease in a model where antigen and adjuvant are abundant.

Therefore, it has been suggested that the Abstract, Discussion, as well as the title of your manuscript be revised. The changes should reflect the fact that you have not demonstrated that the reason for the diminished EAE is indeed the reduction of naive T cells (unless you can, in fact, provide such data, ruling out alternative interpretations).

As mentioned above the transfer of naïve 2D2 CD4 T cells into post-septic hosts demonstrates that the post-septic environment is not constraining CD4 T cells in their capacity to survive, respond to antigen, and elicit disease (Figures 6 and 8). Additionally, the competitive expansion in Figure 7 demonstrates that sepsis does not intrinsically impair CD4 T cell expansion. However, given that we cannot completely rule out all other hypotheses we have modified the text to reflect this request.

There are other specific questions that arose given the interpretation of the current data. The authors show the group of mice that received 2D2 T cells prior to CLP showed a similar disease reduction as mice that are only immunized without receiving any additional T cells (Figure 1). Even if the number of originally transferred 2D2s (5x10^6^) was reduced after CLP compared to the post-CLP transfer (Figure 6—figure supplement 2), one would still expect to have a significantly increased number of MOG-specific T cells in the system compared to mice that did not receive any T cells. This then casts doubt on the fact that the reduced number of naïve T cells in post-septic mice is the reason for the EAE amelioration.

This comment does not accurately describe the experiment we performed. No mice that were immunized to induce EAE received 5x10^6^ cells. The number of naïve 2D2 CD4 T cells that were transferred was 1000-fold less than the reviewer states. We only transferred 5x10^3^ naïve 2D2 CD4 T cells into mice that received immunization. The only group of mice receiving 5x10^6^ 2D2 CD4 T cells were those that were non-immunized in Figure 7 to assess the relative survival capacity of 2D2 CD4 T cells derived from Sham and CLP hosts.

Why is 5x10^3^ naïve CD4 T cells close to normal physiology?

Prior work demonstrated there are approximately 1000 naïve MOG-specific cells within a C57BL/6 mouse (Martinez, RJ, et al. Nat. Commun. 2016 7:13848). Further, our years of experience with adoptive transfer of naïve T cells suggests that the majority of cells do not survive these transfers, with a general notion that only 10% of the transferred cells successfully “take” (i.e., engraft) (please refer to our Immunity Resource article 2007 – 26:847 for further explanations of the TCR-Tg T cell adoptive transfer models). Thus, we anticipate approximately 500 naïve 2D2 CD4 T cells to engraft in the hosts given 5x10^3^ cells. Therefore, we do not view the transferred 2D2 CD4 T cells as dramatically increasing the number of naïve MOG-specific cells within the host as the reviewer suggested.

The authors alternatively suggest a potential priming deficit as an explanation for their phenotype, however, they do not follow up on this. They indeed see a reduced frequency of MOG-specific T cells in the draining lymph nodes at day 7 post immunization (Figure 4), but do not explain a potential mechanism.

We addressed the capacity of CD4 T cells to be primed in the draining lymph node in Figure 6, where the post-transfer cells were not limited in their numerical expansion in the lymph node, nor were they impaired in proliferative capacity or by increased cell death. Additional text has been added to further clarify this point.

1) Figure 1A the very first finding to start off the whole story, shows only up to day 13. This is way too short for evaluating EAE and certainly at the suboptimal quality of its experimental setup. Because of it, EAE with CLP may be just simply delayed, and the authors cannot claim the EAE "ablated" and "Reduction in disease severity."

We performed additional experiments extending the duration of EAE and continued to observe consistently diminished EAE disease in CLP hosts (up to 40 days after immunization) (new Figure 1—figure supplement 1). Thus, our data demonstrate that EAE is ablated in CLP hosts.

2) Why was CD11a used to gate on T cells? Normally one would gate on a T cell marker such as CD90 or just CD4.

Why use CD11a?

CD11a is ubiquitously expressed on T cells. Thus, it is superior to CD4 alone in the identification of CD4 T cells, and permits us to exclude CD4-expressing non-T cells.

Numerous publications have demonstrated that increased CD11a expression can aid in the identification of antigen-experienced CD4 and CD8 T cells (McDermott, DS, et al. J Immunol 2011 187(11), Rai, D, et al. J Immunol 2009 183(12), Butler, NS, et al. Nat Immunol 2011 13(2)). To date, since our initial description of CD11a as a useful discriminator of those T cells that have responded to cognate Ag and those that did not (naïve), more than 50 groups have adopted this idea to track Ag-specific T cells in response to more than 20 different pathogens/immunizations. Therefore, CD11a is superior to CD90, whose expression is not modulated following antigen stimulation. An example of this utility can be observed in Figure 2E where MOG-specific cells have increased expression of CD11a relative to non-MOG-specific CD4 T cells. Thus, increased expression of CD11a on the MOG-specific CD4 T cells served as an internal assurance of bona fide tetramer^+^ MOG-specific CD4 T cells that had encountered antigen.

3) In Figure 3, please show a dot plot for each case (Sham and CLP) for IFNγ versus IL-17A and one of the two versus TNFa, or even better, GM-CSF versus TNFa. FMO can be shown in the Supplementary figure showing the data as histograms is not informative.

We have modified Figure 3 to display dot plots. FMO is now displayed in new Figure 3—figure supplement 1.

4) What is the gate used in Figure 6E?

As stated apoptotic cells can be identified by presence of active caspase 3/7 (i.e., FLICA^+^) and membrane depolarization (i.e., propidium iodide [PI^+^]). However, given the reviewer’s concern additional clarification of this point has been added to the text and the figure legend for Figure 6.